



# Stability of the Regional Brewer Calibration Center for Europe Triad during the period 2005 – 2016

Sergio Fabián León-Luis[1,2], Alberto Redondas[1,2], Virgilio Carreño[1,2], Javier López-Solano[1,2,3], Alberto Berjón[2,3], Bentorey Hernández-Cruz[1,2,3], and Daniel Santana-Díaz[2,3]

[1]Izaña Atmospheric Research Center, Agencia Estatal de Meteorología, Tenerife, Spain
[2]Regional Brewer Calibration Center for Europe, Izaña Atmospheric Research Center, Tenerife, Spain
[3]Departamento de Ingeniería Industrial, Universidad de La Laguna, Tenerife, Spain

*Correspondence to:* Alberto Redondas (aredondasm@aemet.es)

**Abstract.** Total ozone column can be measured using Brewer sprectrophotometers which are calibrated periodically in inter-comparison campaigns with respect to a reference instrument. In 2003 the Regional Brewer Calibration Centre for Europe (RBCC-E) was established at the Izaña Atmospheric Research Centre (Canary Islands, Spain) and from 2011 it has transferred its own calibration mainly to other European Brewers using the Brewer #185 as reference instrument. The RBCC-E organizes

regular inter-comparisons which are held annually alternating between Arosa (Switzerland) and El Arenosillo (Spain). This work is focused on showing the stability of the measurements of the RBCC-E Triad (Brewers #157, #183 and #185) made in the Izaña Atmospheric Observatory during the period 2005 – 2016. In order to study the long-term precision of the RBCC-E Triad, it must be taken into account that each Brewer performs a large number of measurements every day and, hence, it becomes necessary to calculate a representative value of all of them. This value was calculated from methods previously used to

study the long-term behaviour of the World Reference and Arosa Triads. Applying their procedures in our triad allows us to compare the three instruments. In this way, the difference between the values calculated for each Brewer and the triad mean was analyzed. In daily averages, applying the procedure used for the World Triad Reference, the RBCC-E Triad presents a relative standard deviation mean equal to 0.41% ($\sigma_{157} = 0.362\%$, $\sigma_{183} = 0.453\%$ and $\sigma_{185} = 0.428\%$). In opposite, using the procedure of the Arosa Triad, the RBCC-E presents a relative standard deviation around at $\sigma = 0.5\%$. In monthly averages, the

method of the World Triad Reference give a relative standard deviation of 0.33%, 0.34% and 0.23% for Brewers #157, #183 and #185, respectively (0.3% in mean). Whereas, the procedure of the Arosa Triad gives a monthly values 0.3%. In this work, two ozone datasets are analyzed: the first included all the ozone measurements available while the second only includes the simultaneous measurements of all three instruments. Furthermore, in this paper we also describe the Langley method used in the RBCC-E Triad to calculate the Extra-terrestrial constant (ETC), which is the necessary first step to ozone retrieval. Finally,

the short-term, or intraday, stability is also studied to identify the effect of the solar zenith angle on the accuracy of the RBCC-E Triad.



## 1   Introduction.

The ozone layer is a region of the Earth's stratosphere that absorbs most of the Sun's Ultraviolet (UV) radiation. Until a few decades ago, it was thought that the ozone concentration was constant in the stratosphere. However, after the discovery of the hole in the ozone layer in the mid-1980s, this idea was discarded (Farman et al., 1985). The negative impact that ultraviolet radiation has in terrestrial life led to the signing of the Montreal Protocol in 1987, where several countries agreed to reduce the agents that produce this decrease (Sarma and Andersen, 2011). From this date, the monitoring and control of the ozone layer has been a priority of the World Meteorological Organization (WMO). This task requires instruments that can measure the total ozone column concentration with an accuracy around 1-3% such as the Dobson and Brewer which are considered as reference instruments (Basher, 1985; varotsos and Cracknell, 1994; Scarnato et al., 2009; Fioletov, 2005).

Brewer spectrophotometers are widely used to measure the total ozone column (TOC), ultraviolet irradiance and, more recently, the aerosol optical depth in the ultraviolet range (Carvalho and Henriques, 2000; Gröbner et al., 2001; Fioletov, 2002; López-Solano et al., 2017). The Brewer is a spectrophotometer mounted on a sun tracker that determines the ozone concentration from a direct measurement of the solar radiation. For this, each Brewer has a holographic grating system for wavelenght separation and a slit mask mechanism to select the different UV spectral lines to be measured. These lines are associated with maximum and minimum ozone absorption bands.

Although the first Brewer was developed in the early 1980s (Brewer, 1973; Kerr et al., 1985), it has had continuous technical improvements to gain accuracy. This includes improvements in the photomultiplier, diffraction gratings, and operating software, and also the incorporation of new measurement routines (Fioletov et al., 2011; Karppinen et al., 2015; Fountoulakis et al., 2017). However, possibly the greatest improvement has been the transition from a single to a double monochromator. This virtually eliminates the presence of stray light in the measurements that causes a decrease in the TOC concentration at large solar zenith angles (Karppinen et al., 2015). In practice, for angles greater than 70 degrees, single brewers presents this problem. Although, depending on the instrument, its influence may be greater or lesser (Redondas et al., 2015; Redondas and J. Rodriguez-Franco, 2016; Redondas and Rodríguez-Franco, 2015).

The calibration of the Brewer is traceable to the Triad belonging to Environment and Climate Change Canada, consisting in Brewers #008, #014 and #015, which is considered the World Triad Reference. These single Brewers are calibrated every few years at the Mauna Loa Observatory (Hawaii) using the Langley method (Fioletov, 2005). A second triad formed by double Brewers #145, #187 and #191 is also operated in parallel to World Triad Reference in Toronto (Netcheva, 2014; Zhao et al., 2016). Also, the Swiss Federal Office of Meteorology and Climatology (Meteo Swiss) has the Arosa Triad formed by the singles (#040 and #072) and double (#156) Brewers (Stübi et al., 2017). However, the Brewers distributed around the World are calibrated from the travelling standard reference, Brewer #017, managed by International Ozone Services (IOS) and Brewer #158 managed by Kipp & Zonen.

In addition, since November 2003 and within the World Meteorological Organization (WMO) and the Global Atmosphere Watch (GAW) Programme, the Regional Brewer Calibration Centre (RBCC-E) for RA-VI Region was established at the Izaña Atmospheric Observatory (IZO), managed by the Agencia Estatal de Meteorología (AEMET). The RBCC-E is the European



Brewer Reference and hence can calibrate and transfer its own calibration. Its trajectory started in the year 1997 when the first double Brewer #157 was installed at IZO, running in parallel with a single Brewer #033 for six months. In January 2005, a second double Brewer, the #183, was installed and designated as the travelling reference. The single Brewer #033 was moved to Santa Cruz Meteorological Station (SCO) in December 1997, leaving the RBCC-E with only two instruments. In July 2005,

a third double Brewer #185 was installed. Since that moment, the RBCC-E has been formed by the Brewers #157, #183 and #185. The TOC measured by Regional Primary Reference #157 are sent regularly to different world data servers. The Regional Secondary Reference #183 is used to check new routines. Whereas, the Regional Travelling Reference corresponds to Brewer #185.

The Izaña Atmospheric Observatory is located in the island of Tenerife, on the top of a mountain plateau at 2373 m a.s.l.
The observatory is thus located in the region below the descending branch of the Hadley cell, typically above a stable inversion layer, and on an island far away from any significant industrial activities. This ensures clean air and clear sky conditions around all the year and offers excellent conditions to perform the Langley technique. Each Brewer can be calibrated "in situ" and independently using the Langley plot method and without the need to move them to other locations. Moreover, comparisons with the World Triad Reference are carried out regularly. So, the tractability between the RBCC-E and the World
Triad Reference is checked during the calibration campaigns throughout the travelling references #185 and #017. In this comparison, both instruments agree within 0.5%. This values has been calculated using the measurements range where the Brewer #017 does not present stray light (Redondas et al., 2015; Redondas and J. Rodriguez-Franco, 2016; Redondas and Rodríguez-Franco, 2015)

The RBCC-E also organizes inter-comparisons which are held annually alternating between Arosa (Switzerland) and El
Arenosillo (Spain). Since 2011, more than 150 calibrations have been performed (Redondas et al., 2015; Redondas and J. Rodriguez-Franco, 2016). In these campaigns, the RBCC-E facilitates a new calibration for each instrument. Moreover, in order to obtain an ozone value with better accuracy, the RBCC-E advises on the need to apply some type of correction (normally, standard lamp correction) on the measurements performed by each Brewer in its local station before the campaigns. Aside from regular inter-comparisons, the RBCC-E has carried out other research campaigns supported by the ESA CalVal project. The
NORDIC campaigns, with the objective to study the ozone measurements at high latitudes, and the Absolute Calibration Campaigns performed at IZO with the participation of Brewer and Dobson reference instruments. The participating Brewers and the travelling reference #185 operate with the same schedule throughout these campaigns. The TOC concentrations recorded by the travelling reference #185 are used to calibrate the participating Brewers and also to conduct research works (Redondas et al., 2015; Redondas and J. Rodriguez-Franco, 2016; De La Casinière et al., 2005).

Finally, it should be also mentioned that within the framework of COST Action ES1207, "A European Brewer Network" (EUBREWNET), the RBCC-E and AEMET are developing a dataserver for EUBREWNET (http://rbcce.aemet.es/eubrewnet) which will allow to calculate the TOC concentration in near real time (Rimmer et al., 2017). This completes the objectives of this COST action, whose aim is establishing a coherent network of Brewer monitoring stations in order to harmonise operations and develop approaches, practices and protocols to achieve consistency in quality control, quality assurance and coordinated





operations. Currently, around 40 Brewers, mainly European, send their data automatically every 20 minutes to EUBREWNET's dataserver. This dataserver also allows the reprocessing of data for homogenization and automatic quality control.

The present work focused on investigating how the precision between the measurements performed at IZO by the Brewers #157, #183 and #185 every day has evolved over the years to identify periods with lower or higher agreement between the

5 Brewers. In order to compare the stability of our triad with the values reported for the World Triad Reference and Arosa Triad. With this idea in mind, this work has been structured as follows: an approach to ozone retrieval and Langley method is presented in Section 2. The ozone values recorded in the period 2005-2016 and datasets used are shown in Section 3. The methods used to calculate the daily ozone value and the results obtained from these values and its discussion are presented in Section 4. Also, in this section is studied the behavior of the RBCC-E Triad in function on SZA range where the measurements

are performed. Finally, the conclusions are presented in Section 5.

## 2 Theoretical Approach.

### 2.1 Ozone retrieval.

Each Brewer, in its movement following the Sun, measures the direct solar radiation in four spectral lines which are associated with maximum and minimum ozone absorption bands. The line intensity ($F$) can be expressed in terms of counts per second,

after applying some instrumental corrections on the raw counts (the so called dark counts, dead time, and temperature corrections) and also taking into account the contribution of the Rayleigh scattering. The main reason why it is necessary to measure more than one line is the overlapping of the $O_3$ and $SO_2$ absorption bands. So, it is necessary to measure more than one line to get information about the $SO_2$ contribution and subtract it.

The Lambert-Beer's law relates the irradiance at the ground and the top of the atmosphere ($F_i$ and $F_i^0$, respectively) for each

20 different wavelengths (denoted by $i$ subindex) as follows:

$$F_i = F_i^0 e^{-\tau \alpha_i m} \tag{1}$$

where $\tau$ is the gas concentration ($O_3$, $SO_2$, etc) and its absorption coefficient ($\alpha_i$). The distance traveled by the solar radiation is given by the air mass, $m$. Using standard Brewer operational variables, Eq. 1 can be solved for the TOC as follows,

$$O_3 = \frac{MS9 - ETC}{\alpha \cdot m} \tag{2}$$

where ETC and MS9 represent the solar radiation measured at the top of the atmosphere and the ground, respectively (Brewer, 1973; Kerr et al., 1981, 1985; Kipp & Zonen, 2008). Both parameters, and also the ozone absorption coefficient, $\alpha$, must be understood as values obtained from the weighted linear combination of the intensities $F$, in logarithmic scale, of the four lines measured:

$$MS9 = 10^4 \sum_i w_i \log F_i = 10^4 (2.2 \log F_5 + 0.5 \log F_4 - 1.7 \log F_6 - \log F_3) \tag{3}$$



$$ETC = 10^4 \sum_i w_i \log F_i^0 = 10^4 (2.2 \log F_5^0 + 0.5 \log F_4^0 - 1.7 \log F_6^0 - \log F_3^0) \tag{4}$$

$$\alpha = 10^4 \sum_i w_i \log \alpha_i = 10^4 (2.2 \log \alpha_5 + 0.5 \log \alpha_4 - 1.7 \log \alpha_6 - \log \alpha_3) \tag{5}$$

The absorption intensity lines $\alpha_i$ are calculated from dispersion test (Redondas et al., 2014). The weights, $w_i = (-1, 0.5, 2.2, -1.7)$ and the wavelengths used have been especially selected to suppress the aerosol and $SO_2$ effects in the measured signal (Dobson, 1957; Kerr et al., 1981). These $\lambda$ and $w_i$ fulfill the equations 6 and 7.

$$\sum_{i=1} w_i = 0 \tag{6}$$

$$\sum_{i=1} w_i \lambda_i \approx 0 \tag{7}$$

ensuring that any linear effects with wavelength are suppressed and also allow to minimize any small shift in wavelength and the influence of sulfur dioxide on the ozone retrieval.

It is important to note that the factor $10^4$ introduced in Eq. 3 is because the Brewer algorithm works in an internal base 10 logarithmic space multiplied by this factor. Also note that the standard (so-called DS) routine used to determine the ozone concentration from direct sunlight radiation measures the signal intensity in seven slits, $i = 0$–$6$, the second devoted to the dark counts and the remaining ones to the $SO_2$ and $O_3$ absorption. This explains the values of the subscripts in Eq. 3. There is a further slit 0, which is used to measure the spectral line of an internal HG lamp as an auto-calibration method. Slit 2 only contains contributions from the $SO_2$ absorption and is not used in Eq. 3. A more extensive description about dispersion test and the mathematical procedure to calculate the ozone concentration can be found in Kipp & Zonen (2008); Gröbner et al. (1998).

## 2.2 Langley calibration method for the RBCC-E Triad.

All the parameters in Eq. 2 are known except the solar radiation at the top of the atmosphere $ETC$, which must be estimated. The Langley calibration method is the most popular procedure to estimate it. This technique is based on Lambert-Beer's law (Eq. 1) written as a linear equation with the total optical air mass $m$ as the independent variable and $\log_e F_i^0$ as the intercept:

$$\log_e F_i = -\tau \alpha_i m + \log_e F_i^0 \tag{8}$$

This procedure could be applied directly to calculate the solar extraterrestrial constant, in counts per second, for each one of the spectral lines measured, see Eq. 3. In practice, with respect to ozone measurements, the ETC is calculated directly fitting a linear equation to the $MS9$ values respect to air mass $m$,

$$MS9 = ETC + O_3 \alpha m \tag{9}$$





Although, in principle, all measurements could be used for this calculation, experience suggests that is better to select a subset of measurements. This is because the atmospheric conditions are not stable throughout a day and this variability affects the ETC value calculated. Therefore, in practice, the days with a stable atmosphere are selected. Redondas (2008) have shown that the following criteria, listed in order of application, can be used to get a good agreement between the ETC values calculated:

1. Days with a stable ozone concentration (standard deviation lower than 2 DU) and a large number of measurements (more than 50).

2. Days with a low aerosol optical depth concentration and clean-sky conditions.

In this respect, during a large part of the year, the Izaña Atmospheric Observatory presents the ideal atmospheric conditions required to use the Langley calibration method. However, and due to the low latitude of the Canary Islands, the number of
10 measurements performed at low air mass ($1 \leq m \leq 2$) are more than those at large air mass ($m \geq 2$). This produces a non-homogeneous data distribution. For this reason, Redondas (2008) suggest to make a Langley fit in the scale $1/m$. This allows obtaining two ETC values per day. It is important to indicate that despite selecting the better days, when the ETC values obtained in different days are compared, its standard deviation is $\pm$ 5. This difference is considered normal and the ETC introduced in the Eq. 2 corresponds to the ETC mean.

Aside from the interest to determine the ozone concentration, the ETC is considered as a probe to check the correct state of the instrument. So, in a period with a stable behavior, the ETC calculated from the Langley method presents a constant value (std. $\pm$ 5), changing only when the instrument loses its calibration. This may happen, for example, after replacing a damaged component or due to normal drifts by its continued operation. On both cases, and after a stabilization period, a new ETC value can be calculated.

As an example, Fig. 1 shows the operative value of the ETC for the Brewer #183 during the year 2011. The vertical lines represent situations which can produce a change in the behaviour of the instrument, while the horizontal line represents the operative ETC used to calculate the TOC concentration (Eq.2). As it can be observed, the ETC was changed two times, the first time by maintenance tasks (performed by IOS service in July 2011), and the second one due to changes in the Brewer configuration (to be more precise, changes in the so-called "Cal-Step" in August 2011). On the contrary, during the maintenance
tasks (June 2011) or after UV calibration in our facilities (November 2011), the ETC remained constant. Only the Langleys that satisfy the conditions indicated in Sect. 2.2 are used to calculate the weekly mean. Other examples of events that may affect the ETC can be found in the calibration campaign reports (Redondas et al., 2015; Redondas and J. Rodriguez-Franco, 2016).

When a new ETC is given, the TOC concentration calculated from this new configuration can be compared with the values
obtained by other instrument with similar resolution. This is the best strategy to check if the new ETC is the correct one and allows to identify the exact moment when a Brewer begins to have an irregular behaviour and needs a new calibration. At the RBCC-E this task is simple because the Brewers are constantly compared to each other. The tractability between the RBCC-E and the world Reference Triad during the calibration campaigns throughout the travelling references #185 and #017 which is a regular section of the calibration campaign reports.

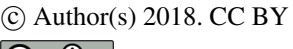



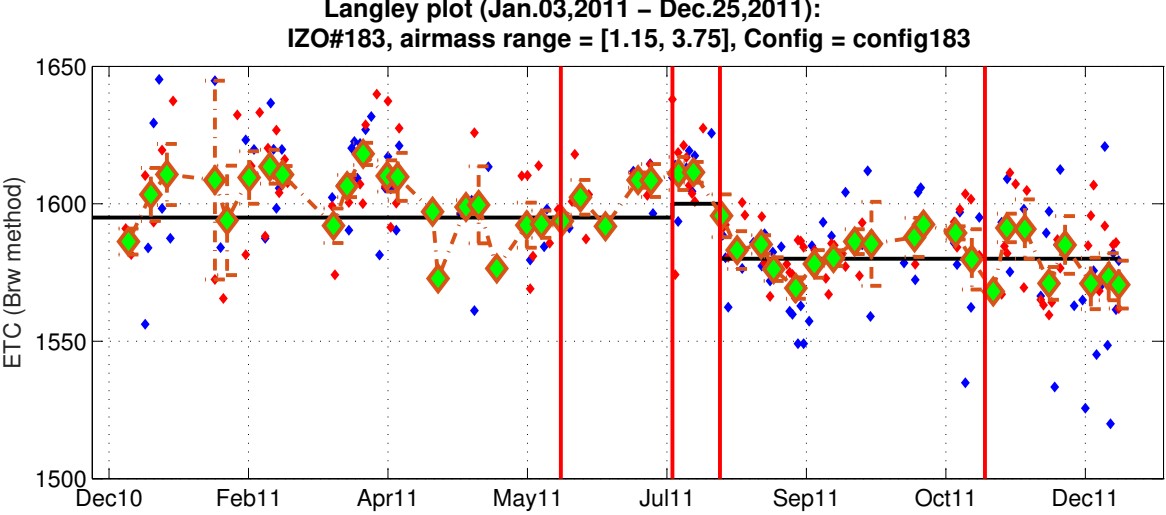

**Figure 1.** Operative ETC value for Brewer #183, after some events that could cause a change in the instrumental calibration during 2011. The red and blue dots denote daily ETC values calculated by the Langley method before and after solar noon, respectively. The diamond symbols and the error bars correspond to the ETC weekly mean and its standard deviation.

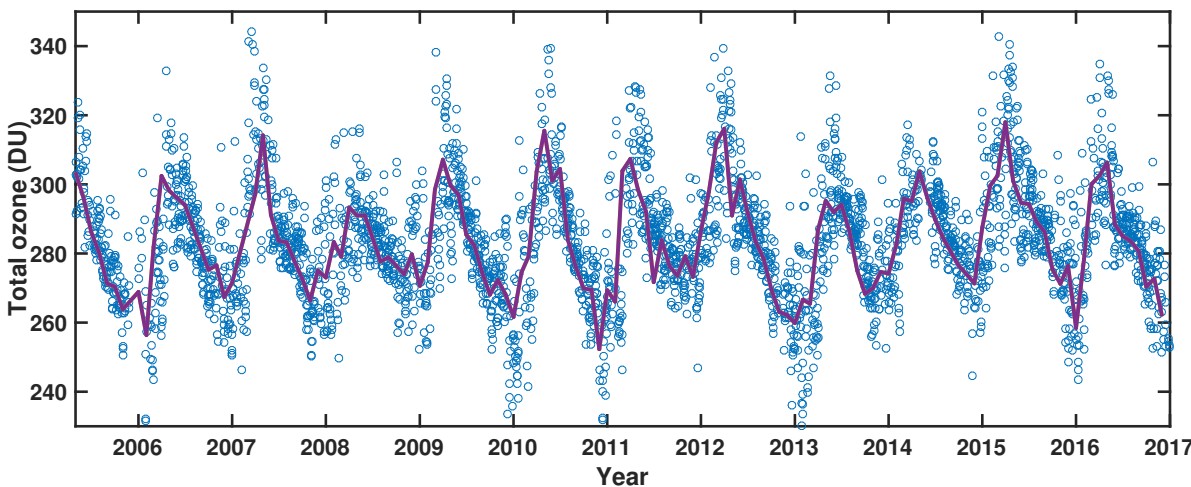

**Figure 2.** Time series of the ozone concentrations measured by Brewer #157 at Izaña Atmospheric Observatory, showing daily (dots) and monthly (line) means.



## 3    Ozone and Dataset selected

In order to summarize the history of the RBCC-E Brewers, Table 1 provides the total number of days and measurements performed by these instruments since they became operational at IZO and as long as the weather conditions allowed them to operate. Fig. 2 shows the daily (circles) and monthly (line) TOC means measured by Brewer #157. As it can be observed, the

ozone presents an annual cycle with a sawtooth profile with maximum and minimum values in spring and autumn, respectively. Despite this annual behaviour, the ozone is stable during the day, with a low standard deviation for the recorded values. This factor, together with a thermal inversion which produces an atmosphere free of anthropogenic pollutants and excellent weather conditions all the year, explain why Izaña Atmospheric Observatory is an excellent location for a Brewer reference centre and, also, why the Langley technique is used as calibration method.

From the times series of the ozone measurements of each brewer in this work two data sets have been built. The first dataset is obtained directly from these times series, after applying several conditions, listed next in order of application:

1. Only include measurements performed at Izaña Atmospheric Observatory.

2. Remove days with problems clearly identified (wrong alignment, etc.).

3. Only the days where the three Brewers have performed measurements are considered in this work. Moreover, each

Brewer must have more than 12 measurements, with a minimum of 4 before and after the solar noon (homogeneous distribution).

4. The standard deviation of the measurements performed by each Brewer must be lower than 0.6. This is introduced to avoid days where the ozone presents a an unusual behaviour.

The second dataset is obtained with the same conditions but also imposing that the measurements must be simultaneous

(condition 3 above). A measurement by a Brewer is considered simultaneous if it is within 5 minutes of measurements performed by the other two Brewers of the RBCC-E Triad. Therefore, this second dataset can be considered a subset of the first. Table 2 gives a summary of both datasets. The entry "Evaluated Days" denotes the number of days used in each dataset to study the stability of the RBCC-E Triad. It is important to note that the Dataset 1 includes the measurements made in the period 2005-2016 while Dataset 2 only considered simultaneous measurements from 2010 onwards. Before 2010, a large part of the

measurement time was focused on the UV spectrum, and, hence, there are fewer ozone direct sun in the instrument schedule. This means that the likelihood of finding 12 simultaneous measurements between the three instruments is low, particularly in winter where the presence of clouds is higher. After 2010, The RBCC-E started using the same synchronization schedule in their Brewers. These schedules take into account the sunrise and sunset times of each day. Therefore, there is one for each day of the year. The routines introduced in it are distributed in function of the solar zenith angle.





**Table 1.** Number of operational days and measurements since their setup for the Brewers of the RBCC-E Triad.

|  | Brewer #157 | Brewer #183 | Brewer #185 |
| --- | --- | --- | --- |
| Operational Days | 3173 | 2740 | 2780 |
| Operational Measurements | 259534 | 204022 | 229201 |

**Table 2.** Summary of the datasets used in this work.

|  | Dataset 1 | Dataset 2 |
| --- | --- | --- |
| Evaluated Days | 2073 | 1325 |
| Period studied | 2005–2016 | 2010–2016 |

## 4   Results and discussion.

The precision of the measurements carried out by the RBCC-E Triad was evaluated from the datasets described in Sect. 3. In the case of the long-term behaviour, it was studied using both datasets, while the short-term behaviour was analyzed using only Dataset 1. The results obtained are shown in statistical terms.

### 5  4.1   Representative value of the Total ozone column.

To our knowledge, there are only a few publications where the stability of the World Reference and Arosa Triads is analyzed. In these articles, and due to the large number of ozone measurements performed throughout day, the authors have calculated a representative value of all of them and, from it, the long-term stability of its triad has been analyzed (Fioletov, 2005; Stübi et al., 2017; Scarnato et al., 2010).

Fioletov (2005) studied the long-term stability of the World Triad Reference in the period 1985 – 2003. In this work, the authors proposed to fit the measurements performed by each Brewer (#008, #014 and #015) to a 2$^{nd}$ grade polynomial (see Fig.3):

$$O_3 = A + B \cdot (t - t_0) + C \cdot (t - t_0)^2 \tag{10}$$

where $O_3$ are the ozone concentrations measured and $t - t_0$ corresponds to the difference between the time of the measurement
and the solar noon. The independent coefficient $A$ obtained through the adjustment is used as a representative ozone value of each instrument. The difference between this coefficient for each Brewer and the Triad mean represents the drifts of each instrument. The stability is studied from daily and monthly mean of these differences. In this work, the results using a 3$^{rd}$ grade polynomial have also been investigated, see Fig.3

Stübi et al. (2017) studied the long-term stability of the Arosa Triad in the period 1988 – 2015. In this study, the authors
considered that the behaviour of the Triad is the most appropriate reference for each day. Therefore, all the measurements made





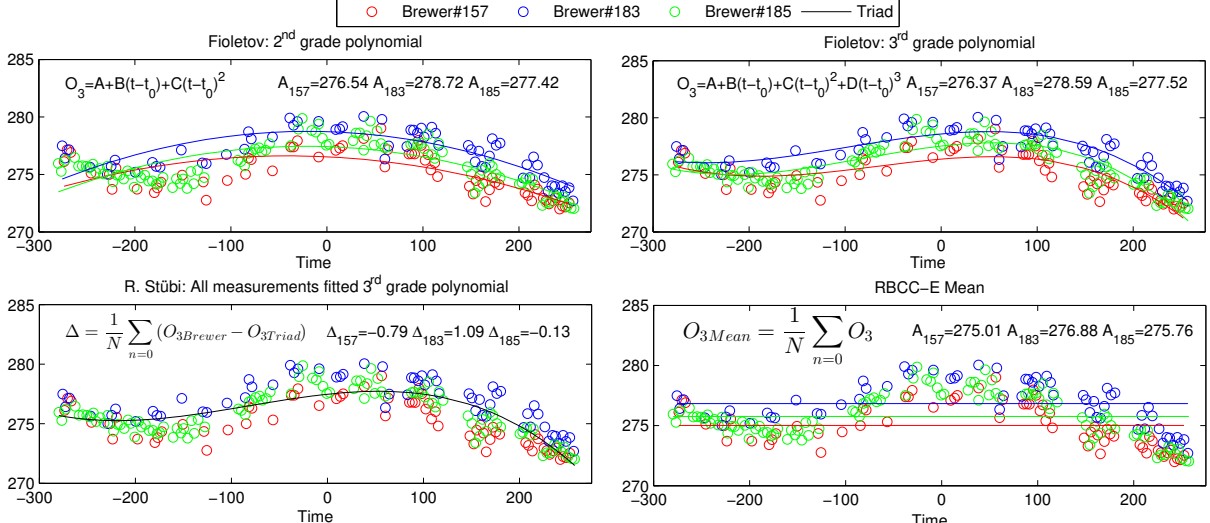

**Figure 3.** Method used to calculated the daily representative value. The ozone values plotted were measured the $16^{th}$ November 2016.

by the three Brewers are modeled as a 3$^{rd}$ grade polynomial dependent on time which represents to the Triad (see Fig.3):

$$O_3 = A + B \cdot (t - t_0) + C \cdot (t - t_0)^2 + D \cdot (t - t_0)^3 \tag{11}$$

where $t_0$ corresponds to the 12 UTC time. In this case, each Brewer is characterized by a shift $\Delta$, which is the mean of the difference between the values measured and obtained from the fit, and a standard deviation $\sigma$. The standard deviation $\sigma$

evaluates the dispersion of these differences. Both parameters are used to analyzed the long-term stability of the Arosa Triad.

In order to compare the long-term stability of the RBCC-E Triad with respect to the World Reference and Arosa Triads, both methods are used to fit our measurements. However, in this work, the time reference $t_0$ is the solar noon.

Although Eqs. 10 and 11 are valid to model the behavior of ozone, it should be noted that the presence of slight mechanical miscalculations in the instrument can affect the final value of the adjustment. Given this problem, calculating the daily mean

of the measurements can be a good strategy to avoid this inconvenient. In this work, and knowing that ozone presents a stable behaviour throughout the day at tropical latitudes, the mean of all measurement of each Brewer was used to calculate a representative value for them, see Fig.3.

$$A_M = \sum_i O_3 \tag{12}$$

The difference between value obtained for each Brewer and the Triad mean represents the drifts of each instrument. Although

the median can be other possibility to study the behaviour of the RBCC-E Triad, our experience suggest that the mean is robust enough. Moreover, since 2003, the mean has been used to detected when one of our Brewer loses its calibration, therefore, it has been interesting to include it in this work. At this point, it is important to note that for the World Triad Reference as for the RBCC-E the representative value of each instrument is calculated, directly, from their measurements. In opposite, in the Arosa





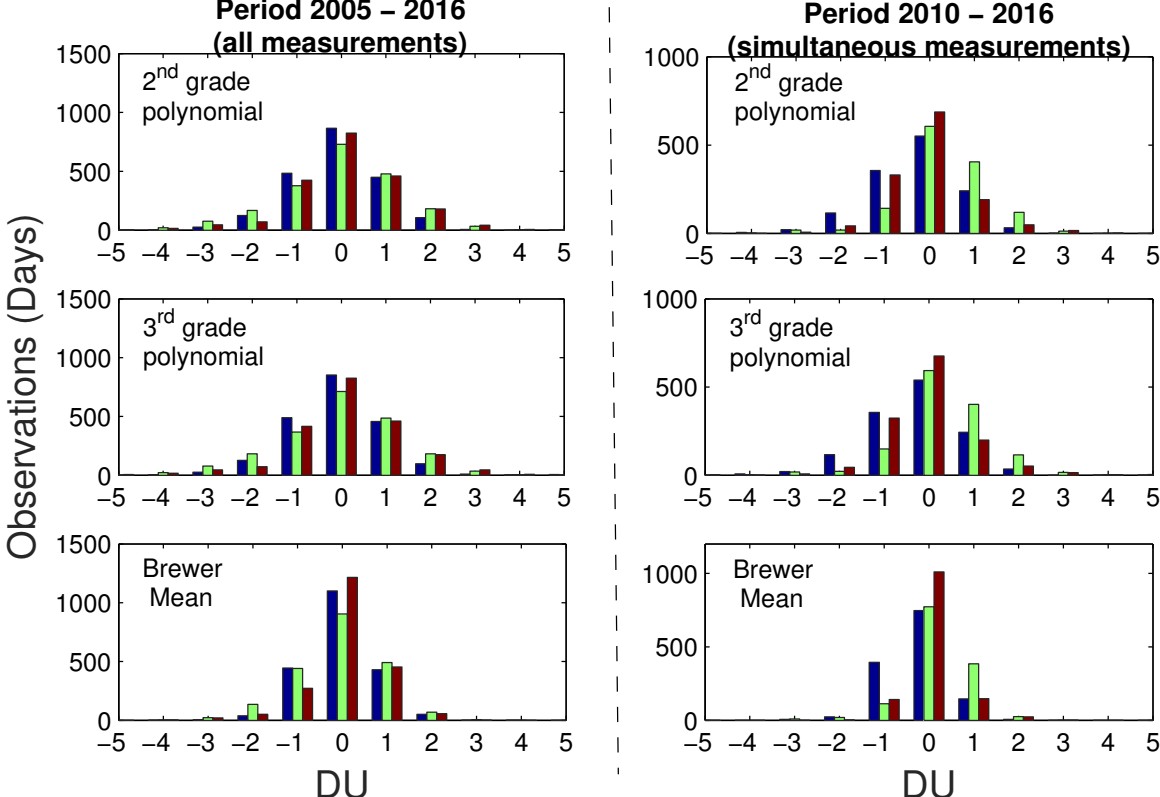

**Figure 4.** Daily difference of the ozone reference value for each individual Brewer with respect to Triad. The values were obtained from the procedure proposed by Fioletov (World Triad Reference) and by daily mean (RBCC-E).

Triad, the representative value of each instrument (denoted as shift $\Delta$) is calculated with respect to the behavior of the three instruments, obtained by adjusting to a polynomial of the third degree (see Fig.3).

### 4.1.1 Long-term stability: Daily averages.

Following the procedure described by Fioletov (2005) for the World Triad Reference, Datasets 1 and 2 (see Sect. 3) were fitted by a 2nd and 3rd grade polynomial. The distribution of the daily difference between the A value, see Eq.10, obtained for each Brewer with respect to the Triad mean are plotted in Fig. 4. Also, in this plot, the difference calculate from daily mean of each instrument was included, see Eq.12.

It is important to take into account that the individual coefficients obtained for each Brewer, and also the Triad mean calculated from them, depend on the method used. As Fig.4 shows, the histograms that represent the results obtained after applying a polynomial fit are similar regardless of the dataset in contrast to the mean. This can be explained by the small variation of the ozone at tropical latitudes. Therefore, the 2nd and 3rd grade polynomial fit give very similar A coefficients, see Fig.3. Therefore,



**Table 3.** Absolute and relative values of the mean shift and the standard deviation.

| | Dataset 1 | | | | | |
|---|---|---|---|---|---|---|
| | $2^{nd}$ grade polynomial (DU) | $3^{rd}$ grade polynomial (DU) | Brewer Mean (DU) | $2^{nd}$ grade polynomial (%) | $3^{rd}$ grade polynomial (%) | Brewer Mean (%) |
| Brewer #157 | $0.787 \pm 1.04$ | $0.796 \pm 1.07$ | $0.569 \pm 0.744$ | $0.276 \pm 0.362$ | $0.279 \pm 0.372$ | $0.1994 \pm 0.26$ |
| Brewer #183 | $0.989 \pm 1.29$ | $1.01 \pm 1.31$ | $0.741 \pm 0.956$ | $0.349 \pm 0.453$ | $0.356 \pm 0.463$ | $0.26 \pm 0.337$ |
| Brewer #185 | $0.89 \pm 1.21$ | $0.90 \pm 1.23$ | $0.56 \pm 0.78$ | $0.315 \pm 0.428$ | $0.32 \pm 0.438$ | $0.20 \pm 0.279$ |

| | Dataset 2 | | | | | |
|---|---|---|---|---|---|---|
| | $2^{nd}$ grade polynomial (DU) | $3^{rd}$ grade polynomial (DU) | Brewer Mean (DU) | $2^{nd}$ grade polynomial (%) | $3^{rd}$ grade polynomial (%) | Brewer Mean (%) |
| Brewer #157 | $0.795 \pm 1.00$ | $0.82 \pm 1.05$ | $0.534 \pm 0.661$ | $0.278 \pm 0.349$ | $0.286 \pm 0.368$ | $0.186 \pm 0.23$ |
| Brewer #183 | $0.747 \pm 0.942$ | $0.784 \pm 1.02$ | $0.547 \pm 0.719$ | $0.262 \pm 0.331$ | $0.275 \pm 0.36$ | $0.192 \pm 0.254$ |
| Brewer #185 | $0.64 \pm 0.87$ | $0.67 \pm 0.99$ | $0.376 \pm 0.526$ | $0.227 \pm 0.311$ | $0.238 \pm 0.333$ | $0.133 \pm 0.189$ |

the method selected to evaluate the precision of the measurements made by the Brewers plays an important role, because the Brewer-Triad mean differences are directly associated with it. In this case, it may be more appropriate to use the daily mean to evaluate the RBCC-E Triad. Regardless, independently of the method used, Fig. 4 shows that, for the great majority of days, the Brewers present less than 2 DU of difference with respect to the Triad mean. This result suggests that the Brewers of the

5 RBCC-E Triad are in good agreement among themselves.

Using the same procedure to evaluate the long-term behaviour can be the best strategy to compare different triads to each other. In this sense, Fioletov (2005) only reported as daily data that the standard deviation mean of the World Triad Reference is equal to 0.47%. This value is the average of the relative standard deviations of each Brewer. Table 3 contains the difference mean, calculated from the mean Brewer-Triad difference plotted in Fig. 4, and its standard deviation. The RBCC-E Triad

presents a relative standard deviation mean equal to 0.41% ($\sigma_{157} = 0.362\%$, $\sigma_{183} = 0.453\%$ and $\sigma_{185} = 0.428\%$; see Table 3, Dataset 1, $4^{th}$ column). This result indicates that the dispersion of the measurements of the RBCC-E Brewers presents the same behaviour as those of the World Triad. Furthermore, the standard deviation values obtained confirm that the daily mean is the best method to evaluate the RBCC-E Triad.

In order to compare the daily behaviour of the Arosa and RBCC-E Triads, a $3^{rd}$ grade polynomial was fitted to all the daily

measurements made by RBCC-E Brewers for Datasets 1 and 2. Then, for each Brewer its mean shift, $\Delta$, and standard deviation, $\sigma$, were calculated (Stübi et al., 2017). The values obtained for the Dataset 1 are shown in Fig. 5. Because Brewer #183 was damaged by a storm and was inoperative between December 2005 and September 2006, the data plotted in that period were calculated from measurements of Brewers #157 and #185 only. Similarly, when Brewer #185 is away from IZO in calibration





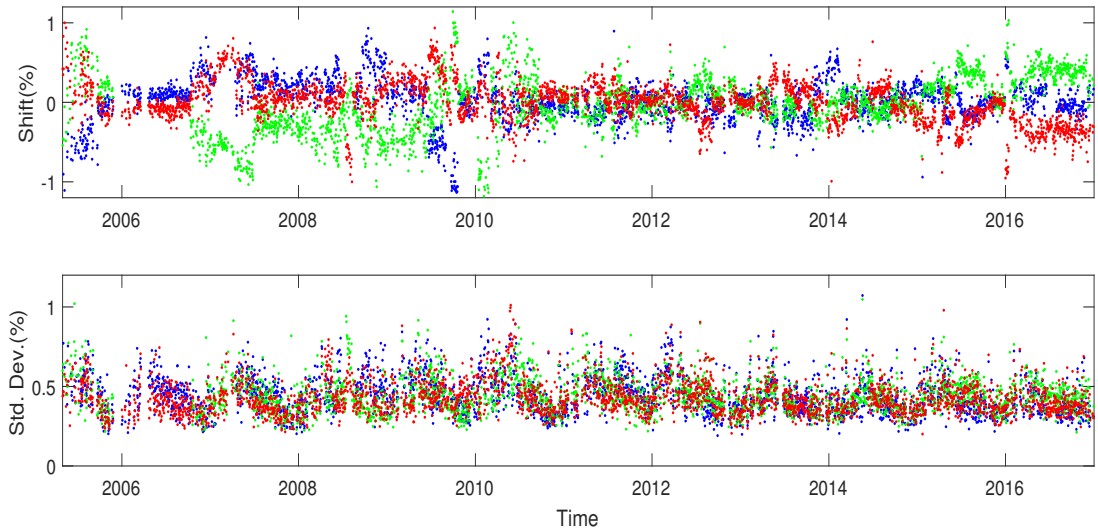

**Figure 5.** Time series of the mean shift $\Delta$ and the standard deviation $\sigma$, in terms relative to the TOC calculated from measurements performed by the three Brewers of the RBCC-E Triad (#157, #183 and #185) fitted with a $3^{rd}$ grade polynomial

campaigns, the values plotted correspond to Brewers #157 and #183. Note that although these data were introduced in Fig. 5 to avoid gaps in the plot, they are not considered in the statistical study. Therefore, the dates evaluated correspond with the days when the full RBCC-E Triad is operative, and the criteria established in Sect. 3 are still used.

As it can be observed in Fig. 5, the results obtained for all instruments in Dataset 1 show a ($\pm 0.5$) value for the mean shift.

A similar result was obtained for Dataset 2 (figure not shown). Contrary to the report in Stübi et al. (2017), in the present case the standard deviation does not show any seasonal component. Again, this result is explained by the almost constant value of the ozone at tropical latitudes. For the Brewers of the RBCC-E Triad, the standard deviation is more influenced by any anomalous internal behavior of the instruments. For middle latitudes, e.g. in Arosa, there is a larger daily variation in ozone and the standard deviation shows it.

Following Stübi et al. (2017), the Table 4 shows the distribution of percentiles of the mean shift and the standard deviation values plotted in Fig. 5. The Brewers present a similar interpercentile range $P_{2.5} - P_{97.5}$, with a mean value close to 1.1%. This result is consistent with the standard deviation shown in Table 3 for the polynomial fits, where the long-term stability of the Brewer-Triad mean was studied by the procedure proposed by Fioletov (2005). In comparison with the Arosa Triad, only Brewer #040 shows a better behavior than the RBCC-E Brewers. The other two Brewers of the Arosa Triad (B#072, B#156)

show similar values to those of the RBCC-E





**Table 4.** Percentiles of the difference distribution (%) for the RBCC-E Triad.

| Dataset 1 | | | Shift $\Delta$ | | | Standard Deviation $\sigma$ | | | | |
|---|---|---|---|---|---|---|---|---|---|---|
| Brewer | $P_{2.5}$ | $P_{25}$ | Median | $P_{75}$ | $P_{97.5}$ | Brewer | $P_{2.5}$ | $P_{25}$ | Median | $P_{75}$ | $P_{97.5}$ |
| #157 | -0.426 | -0.13 | 0.017 | 0.138 | 0.464 | #157 | 0.25 | 0.32 | 0.37 | 0.426 | 0.595 |
| #183 | -0.71 | -0.219 | -0.016 | 0.191 | 0.58 | #183 | 0.25 | 0.327 | 0.377 | 0.439 | 0.599 |
| #185 | -0.54 | -0.099 | 0.0155 | 0.147 | 0.51 | #185 | 0.24 | 0.316 | 0.375 | 0.451 | 0.631 |
| Triad | -0.599 | -0.14 | 0.008 | 0.156 | 0.524 | Triad | 0.248 | 0.321 | 0.373 | 0.439 | 0.609 |

| Dataset 2 | | | Shift $\Delta$ | | | Standard Deviation $\sigma$ | | | | |
|---|---|---|---|---|---|---|---|---|---|---|
| Brewer | $P_{2.5}$ | $P_{25}$ | Median | $P_{75}$ | $P_{97.5}$ | Brewer | $P_{2.5}$ | $P_{25}$ | Median | $P_{75}$ | $P_{97.5}$ |
| #157 | -0.573 | -0.265 | -0.053 | 0.084 | 0.365 | #157 | 0.25 | 0.33 | 0.389 | 0.459 | 0.651 |
| #183 | -0.475 | -0.093 | -0.037 | 0.264 | 0.559 | #183 | 0.265 | 0.349 | 0.405 | 0.473 | 0.627 |
| #185 | -0.341 | -0.103 | 0.007 | 0.114 | 0.513 | #185 | 0.247 | 0.33 | 0.389 | 0.479 | 0.6509 |
| Triad | -0.503 | -0.136 | -0.001 | 0.1386 | 0.517 | Triad | 0.242 | 0.33 | 0.38 | 0.464 | 0.639 |

### 4.1.2 Long-term stability: Monthly averages.

Although the histogram and the statistical parameters already presented suggest that the long-term stability of the RBCC-E Triad is similar to that of the World Reference and Arosa Triads, it can be more interesting to study the stability using monthly means. With this idea in mind, the values plotted in Figs. 4 and 5 were averaged. Fig.6 shows, in relative terms, the monthly values for the period 2005–2016 (Dataset 1). The results confirm that the RBCC-E Triad has a good long-term precision, regardless the method selected. In order to compare with the World Triad Reference, the standard deviation of the monthly values calculated from $2^{nd}$ grade polynomial fit was calculated. As reported by Fioletov (2005), the standard deviation of the 3-monthly mean for the World Triad is 0.40%, 0.46% and 0.39% for Brewers #008, #014, and #015, respectively (0.4% in mean). The RBCC-E Brewers have lower monthly values 0.33%, 0.34% and 0.23% for Brewers #157, #183 and #185, respectively (0.3% in mean). This result indicates that RBCC-E Triad present a 40% lower dispersion of their measurements. Furthermore, Stübi et al. (2017) reported that in the period 2004–2012 the Arosa Triad presented monthly shift lower than $\pm 0.4\%$ while the RBCC-E are lower than $\pm 0.3\%$.

### 4.2 Short-term stability.

Dataset 1 was used to study the short-term stability of the RBCC-E Triad, with a view to determine in which SZA range the consistency of the measurements is higher. The measurements made by the three Brewers every day were fitted by $3^{rd}$ grade polynomial as shown in Fig. 7. As previously commented, this polynomial represents the behavior of the triad and allow obtaining the ozone concentration in function on the time. Similarly to the previous study, each Brewer was characterized by



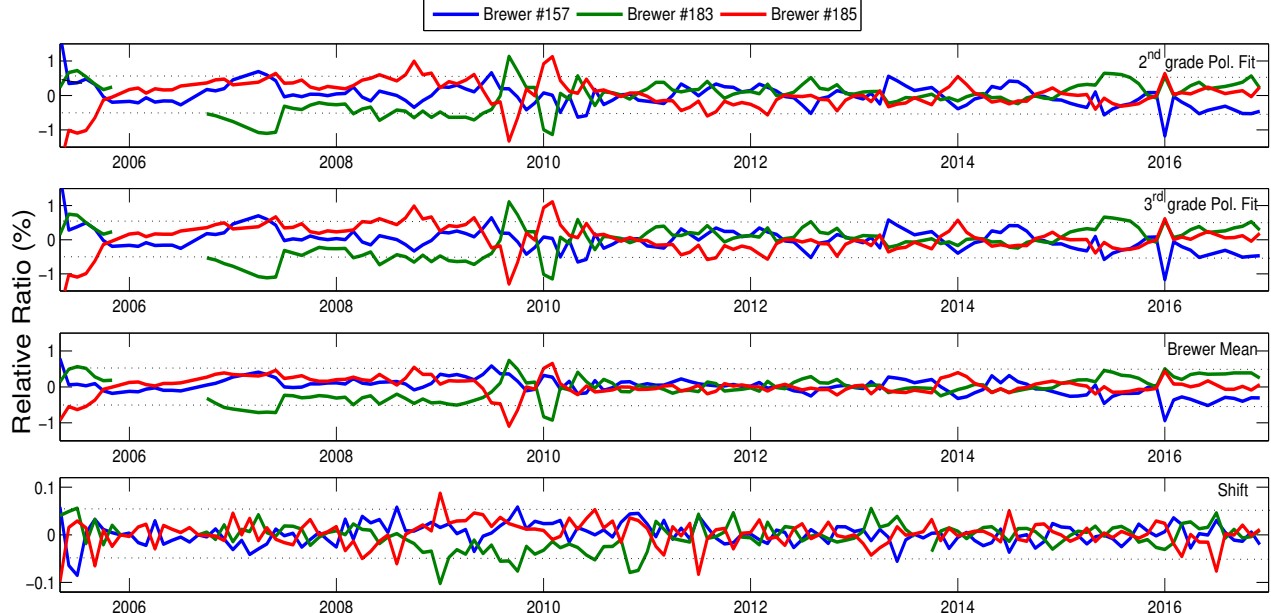

**Figure 6.** Relative ratio of the monthly values with respect to the Triad mean for the method proposed for World Triad Reference, Fioletov (2005), and Arosa Triad, Stübi et al. (2017), and Daily Mean (RBCC-E). The gap for the Brewer #183 data was caused by the tropical storm "Delta" which damaged the instrument.

a shift $\Delta$. In this case, the ozone measurements were divided in function on the SZA. Different SZA ranges were checked, finding that the analysis can be reduced to just three broad ranges:

1. SZA>60º corresponds to the first and last ozone measurements of every day, when solar radiation presents a low intensity and high Rayleigh scattering.

2. SZA<30º corresponds to the measurement in the middle of the day, when the air mass is close to 1 and, hence, there is less Rayleigh scattering.

3. 60º<SZA<30º, the rest of ozone measurements.

In Fig. 7, the box-plot shows the statistical distribution of the mean shift calculated in function on the SZA. As it can be observed, the greatest dispersion values are at low SZA. This indicates that it is at the middle of the day when more discrepancy can be observed between the ozone concentration recorded by the Brewers. This result may seem suprising because in these conditions the solar radiation on the Earth's surface is maximum and the Rayleigh scattering is minimum. This can be easily explained from Eq. 2. In this expression, at low SZA the optical mass is near to 1 and the ozone absorption $\alpha$ is a constant value. Therefore, the denominator take a value almost constant. Thus, a small fluctuation (noise) associated with the MS9



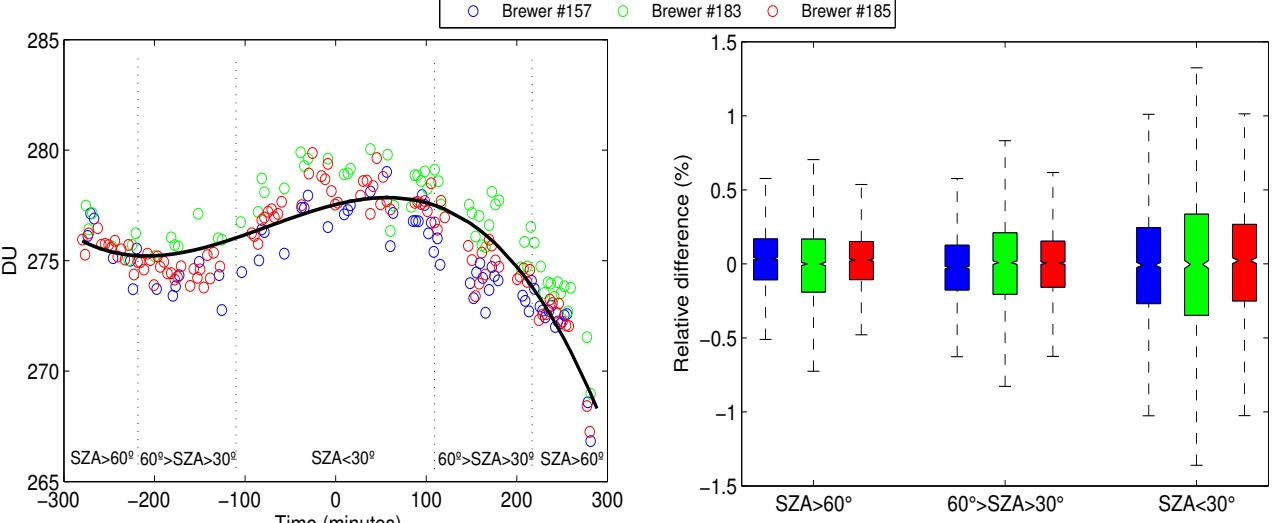

**Figure 7.** Experimental measurements and $3^{rd}$ grade Triad fit (left) and relative difference by Brewer as function of the SZA (right).

values may affect significantly the ozone concentration recorded. In conclusion, the other ranges are where the best stability can be observed.

# 5 Conclusions

The consistency of the ozone measurements from Brewers #157, #183 and #185 of the RBCC-E Triad has been studied in order to check its long-term stability and compare with the behavior reported for the World Reference and Arosa Triads. With this idea in mind, the methods used to study the stability of these triads were reproduced. Based on the procedure used to evaluate the World Triad Reference, the RBCC-E Triad presents a relative daily standard deviation mean equal to 0.41% ($\sigma_{157} = 0.362\%$, $\sigma_{183} = 0.453\%$ and $\sigma_{185} = 0.428\%$), lower than the reported value for the World Triad Reference (0.47 %). Using monthly averages, the standard deviation takes values slightly lower than those obtained from the daily data. Applying the procedure used to study the Arosa Triad, the Brewers of the RBCC-E Triad present a similar interpercentile range $P_{2.5} - P_{97.5}$, with a value close to 1.1%. This value is similar to those reported for the Arosa Brewers, except in the case of Brewer #040. However, the monthly means are better for the RBCC-E Triad with a value lower than $\pm 0.3\%$. The values obtained for the different Triads are fairly similar, see Table 5, which ensures the traceability of the ozone measurements all around the world.

*Competing interests.* The authors declare that they have no conflict of interests.



**Table 5.** Sumary of the three studies compared of the percentage mean standard deviation of the World Triad Reference, Arosa and RBCC-E Triads

|          | three months | daily   |
|----------|--------------|---------|
| Fioletov | 0.42         | 0.47(*) |
| Stuevi   | -            | 0.36    |
| this work| 0.27         | 0.37    |

\* The standard deviation calculated from procedure of Fioletov (2005) and Stübi et al. (2017) are not equal but they are similar enough to compare the three triads

*Acknowledgements.* The authors are grateful to the IZO team and especially all observers and Phd Students who have worked in the past at Izaña Atmospheric Observatory. This article is based upon work from the COST Action ES1207 "The European Brewer Network" (EU-BREWNET), supported by COST (European Cooperation in Science and Technology). This work has been supported by the European Metrology Research Programme (EMRP) within the joint research project ENV59 "Traceability for atmospheric total column ozone" (AT-
5   MOZ). The EMRP is jointly funded by the EMRP participating countries within EURAMET and the European Union.





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
