# Peer review of "Stability of the Regional Brewer Calibration Center for Europe Triad during the period 2005 – 2016"

_Atmospheric Measurement Techniques, 2017_

## Referee Comment (RC1) · Anonymous Referee #1 · 9 Feb 2018

General Comments:

It is surely a very important information for the ozone community how stable the Brewer triad at the RBCC-E is, as these reference Brewers are used for the calibration of a large part of the Brewer network. Unfortunately this paper is in some parts confusing and although I have to commit that English is not my first language, I have the feeling that the text is not written in good English. These issues cause sometimes problems to understand the text. Additionally there are some statements and descriptions which are not correct or at least unclear: SO2 as parameter (additionally to the main parameter TOC, UV and AOD) is in fact mentioned in section 2.1, but not in the correct context.

[Figure]

Brewer spectrometers measure in five wavelengths and not four to get SO2-values too. This parameter should be nowadays close to zero under normal conditions and clean air, but is important in polluted areas and under volcanic plumes. And it is important when a comparison between Dobson and Brewer is investigated.

Another unclear description is the use of airmass m instead of the relative optical path through the ozone layer mu ($\mu$).

The description of the triad system is not profound and clear. Where is the second triad located? In Toronto too? How are this triad and in addition the travelling Brewer No. 17 calibrated? It is good to know how stable the RBCC-E triad is, but what about its accuracy? The agreement between this triad and the World Reference Triad is confirmed by the 0.5% - agreement with #017. But how good is the agreement of #017 with World Reference Triad? When was its last absolute Langley calibration performed?

My recommendation is, that, in addition to the specific comments correction, the paper needs a basic rewriting taking into account the above mentioned shortcomings.

Specific Comments:

- In the entire text: behavior or behaviour should not be mixed. Replace tractability with traceability.

- P1, l9 (abstract): add . . . . . .from "two different" methods previously. . . . . .

- P1, l10: it should already be mentioned here that the World Reference is the Toronto triad.

Introduction:

- P2, l5: "on" instead "in".

- P2, l6: "cause" instead "produce".

- P2, l9 and in references P20, l4: "Varotsos" instead "varotsos".

- P2, l9: "......which are considered to be reference instruments...." Is not correct. The triads or Brewer #17 are references (or standards). May be "basic instruments in the global network" or similar is better.

- P2, l10: "Brewer spectrophotometers are widely used" since when?

- P2, l12/13: "ozone concentration" is not correct, "TOC" is better.

- P2, l16: Why is "although" used? Better "After the development of the first Brewer in the early 1980s it has had continuous......".

- P2, l21: 70 degrees as limit for single Brewer observations seems small, as it corresponds to $\mu$-values of smaller than 3. Even single Brewers can measure reliable TOC up to 75 degrees ($\mu$-values around 3.5) under normally clear air conditions. Perhaps it would be good, to describe the different stray light issues: internal straylight-problem, which is larger for single Brewers, and external stray light (diffuse sky light around the sun), which is similar for all Brewers, but larger for Dobsons. The effect is the same in both cases: drop of TOC, when the SZA ($\mu$) increases and gets large depending on instrument and turbidity.

- P2, l24: The calibration of the Brewer "in the first years" is traceable.

- P2, l31: by the "manufacturer" Kipp & Zonen.

- P3, l5: Add "triad" behind RBCC-E.

- P3, l9: Replace "in the island" with "on the island".

- P3, l16: "These values have been".

- P3, l20: "more than 150 Brewers have been calibrated".

- P4, l5/6: This is not a complete sentence.

Chapter 2 (is in general confusing):

- P4, l13 ff, p5: see under general comments (SO2!). The reference Dobson 1957 is out of place here and I think anywhere in the text. So omit it in the references. The presentation of the corresponding absorption spectra (O3 and SO2) might be helpful.

- P5, l15: DS observations are done in 6, not in 7 slits: 4 ozone, 1 SO2 and 1 dark, the seventh slit is for HG-test.

- P4/5/6: Use of m instead of $\mu$ is not correct.

- P6, l7: "clear" instead of "clean"-sky.

- P6, l11: Why does the use of 1/m (should also be $\mu$) allow obtaining two ETCs? I thought it is the splitting in fore- and afternoon observations.

- P6, l 32-35: no complete sentence.

Chapter 3:

- P8, l6: Giving a number for the low standard deviation would be helpful.

- P8, l9: "Langley technique can be used" instead of "Langley-technique is used".

- P8, l17: The criterion lower than 0.6 under item 4 is not clear: standard deviation of 0.6 DU? Seems to be a very low standard deviation for a day with small ozone variation.

- P8, l18: correct "a an".

- P8, l20: what does "(condition 3 above)" mean in the context with simultaneous? Does it mean, that in addition to condition 3, these selected measurements should be simultaneous?

Chapter 4:

- P10, 4: "mean of daily differences" or "daily mean of the difference"?

- P10,l5: Replace "analyzed" with "analyze".

- P10, l8/9: what does slight mechanical miscalculation mean?

- P10, l 10; replace "inconvenient" with "inconvenience".

- P10, l11 and P11, l11: isn't Izana a subtropical station?

- P10, l15: "another" instead of "other".

- P11, figure 4: The "A" from Eq. 10 is missed. Shouldn't it be placed behind "ozone reference value"? The colors are not assigned to the Brewers.

- P11, last section, P12, first section: Sorry, but I have problems to understand what is meant. The beginning of two consecutive sentence with "Therefore" does not sound good. It is described that the polynomial method show similar results regardless of the data set and the order of the fit. Why is then the daily mean method more appropriate?

- P13, Fig.5: Again the colors are not assigned to the Brewers.

- P13, l5: Why are parentheses used?

- P13, l7: tropical replaced with subtropical.

- P12 – P15: What about Brewer #183? Tables 3 and 4 (Data set 1) and Figures 5 and 6 show a larger ratio in 2007, which cannot be seen in the graph of Arosa method. Only the table 4 gives a negative shift median and larger percentile numbers for Brewer #183 data set 1. Is the reason known? The same is valid for the larger scattering of all RBCC-E Brewers around 2010.

- P14, l9 (and in abstract too): Where do the numbers for the RBCC-E triad come from? I cannot find them in the tables.

- P14, l10: how is 40% lower dispersion determined?

- P14, l12: Is the 0.3% value for RBCC-E also calculated in the 2004 – 2012 period of the Stübi-paper.

- P15, Figure 6: The order of the panels does not coincide with the order of the mentioned methods in the figure caption.

- P15, l10: correct "surprissing".

- P15/16: The explanation for the larger scattering at high sun is good and comprehensible. But the obvious difference of low sun in the forenoon and in the afternoon is neither mentioned nor explained. Is there also an explanation, why Brewer #183 shows larger scattering in the relative difference than the other two Brewers?

- P16/17: The values in table 5 are not clear. Where do they come from? Fioletov daily of 0.47% is mentioned on P16, l8. In the same context the RBCC-E triad number is given as 0.41% in the Conclusion, but as 0.37 in the table, which is confusing.

- P17, table 5: Replace "Stuevi" with "Stübi".

---

## Referee Comment (RC2) · V. Savastiouk (Referee) · 9 Feb 2018

This can be an important paper showing the quality of work these authors are doing at Izana. From personal experience I know that the work done at RBCC-E is excellent. However, this version of the paper is extremely poorly written. Not only it requires a complete overhaul of the English language usage, it also needs some proper structure and logic. If I didn't know the authors personally I'd think they don't understand what they are writing about, to be honest. The Brewer spectrophotometer doesn't "measure spectral lines" as the paper suggests. Saying that "several" countries agreed to the Montreal Protocol is irresponsible, there were 197 countries, including all UN members

and the EU. And the presence of SO2 is not the main reason for measuring at more than one wavelength with the Brewer. I strongly recommend to clearly state the goal of the paper in the abstract and then support that goal in the text. Try to be focused, do not deviate to other (however interesting) topics. Re-arrange the paper to have a logical flow to it. If you introduce a new variable then explain it right-away, not several pages later. The Brewer-related papers from RBCC-E keep using the slit numbering that is not conventional and keep calling the dark count as a slit measurement. I strongly disagree with this terminology. The paper suggests using the Brewer ozone observations as an indicator for selecting the days when ozone is stable for Langley method. It is fundamental to understand that this explicitly says that you assume that the instrument(s) is (are) calibrated already and do not require a new ETC.

Please see the supplement PDF file with detailed comments and highlights/underlines of questionable statements. Looking forward to seeing a new version of this paper.

Please also note the supplement to this comment:
https://www.atmos-meas-tech-discuss.net/amt-2017-460/amt-2017-460-RC2-supplement.pdf

[Figure]

**Supplement:**

[Figure]

**Stability of the Regional Brewer Calibration Center for Europe Triad during the period 2005 – 2016**

Sergio Fabián León-Luis[1,2], Alberto Redondas[1,2], Virgilio Carreño[1,2], Javier López-Solano[1,2,3], Alberto Berjón[2,3], Bentorey Hernández-Cruz[1,2,3], and Daniel Santana-Díaz[2,3]

[1]Izaña Atmospheric Research Center, Agencia Estatal de Meteorología, Tenerife, Spain
[2]Regional Brewer Calibration Center for Europe, Izaña Atmospheric Research Center, Tenerife, Spain
[3]Departamento de Ingeniería Industrial, Universidad de La Laguna, Tenerife, Spain

*Correspondence to:* Alberto Redondas (aredondasm@aemet.es)

**Abstract.** Total ozone column can be measured using Brewer sprectrophotometers which are calibrated periodically in inter-comparison campaigns with respect to a reference instrument. In 2003 the Regional Brewer Calibration Centre for Europe (RBCC-E) was established at the Izaña Atmospheric Research Centre (Canary Islands, Spain) and from 2011 it has transferred^s its own calibration mainly to other European Brewers using the Brewer #185 as reference instrument. The RBCC-E organizes

5  regular inter-comparisons which are held annually alternating between Arosa (Switzerland) and El Arenosillo (Spain). This work is focused on showing the stability of the measurements of the RBCC-E Triad (Brewers #157, #183 and #185) made at the Izaña Atmospheric Observatory during the period 2005 – 2016. In order to study the long-term precision of the RBCC-E Triad, it must be taken into account that each Brewer performs a large number of measurements every day and, hence, it becomes necessary to calculate a representative value of all of them. This value was calculated from methods previously used to

10  study the long-term behaviour of the World Reference and Arosa Triads. Applying their procedures in our triad allows us to compare the three instruments. In this way, the difference between the values calculated for each Brewer and the triad mean was analyzed. In daily averages, applying the procedure used for the World Triad Reference, the RBCC-E Triad presents a relative standard deviation mean equal to 0.41% ($\sigma_{157} = 0.362\%$, $\sigma_{183} = 0.453\%$ and $\sigma_{185} = 0.428\%$). In opposite, using the procedure of the Arosa Triad, the RBCC-E presents a relative standard deviation around at $\sigma = 0.5\%$. In monthly averages, the

15  method of the World Triad Reference give a relative standard deviation of 0.33%, 0.34% and 0.23% for Brewers #157, #183 and #185, respectively (0.3% in mean). Whereas, the procedure of the Arosa Triad gives a monthly values 0.3%. In this work, two ozone datasets are analyzed: the first included all the ozone measurements available while the second only includes the simultaneous measurements of all three instruments. Furthermore, in this paper we also describe the Langley method used in the RBCC-E Triad to calculate the Extra-terrestrial constant (ETC), which is the necessary first step to ozone retrieval. Finally,

20  the short-term, or intraday, stability is also studied to identify the effect of the solar zenith angle on the accuracy of the RBCC-E Triad.

[Figure]

**1 Introduction.**

reference?

The ozone layer is a region of the Earth's stratosphere that absorbs most of the Sun's Ultraviolet (UV) radiation. Until a few

reference?

decades ago, it was thought that the ozone concentration was constant in the stratosphere. However, after the discovery of the

there is no hole in the ozone layer!!! The term is "ozone hole" and it has a definition.

[revised manuscript text omitted]

---

## Referee Comment (RC3) · V. Savastiouk (Referee) · 9 Feb 2018

Although the Brewer #017 is regularly compared with the World Brewer Reference in Toronto it is also absolutely and independently calibrated at the Mauna Loa Observatory in Hawaii, US. Some history of such calibrations can be found at http://www.io3.ca/Calibrations/Brewer/017

---

## Referee Comment (RC4) · T. McElroy (Referee) · 13 Feb 2018

**Stability of the Regional Brewer Calibration Center for Europe Triad during the period 2005 – 2016 Sergio Fabián León-Luis, et al. AMT-2017-460**

**Overview**

This is an important paper of interest to the community monitoring ozone from the ground and from space. It contains information which needs to be retained in the literature for the long-term understanding of the performance of the global ozone observing system. However, it is not publishable in its current state and requires major revisions. Some comments were accumulated in the course of making this assessment and are included herein. Hopefully, these will be of use to the authors and perhaps identify some of the shortcomings in the presentation that are hindering its publication at this time. After revision it will need a further review.

**General comments**

Both "Center" and "Centre" have been used to describe the "Regional Brewer Calibration Center for Europe". One should be chosen consistent with its formal use or the style guide for AMT.

The expression 'lines' is usually reserved for emission and absorption lines, which are very narrow. The word bands seems a better choice in referring to the Brewer measurement wavebands. This has been noted in some places but not all.

There is a tendency throughout to use language loosely and elliptically. For example, one does not compare results to a Brewer, one compares Brewer data or measurements to those of another instrument.

**Detailed comments**

Page numbers listed are the PDF pages

Abstract

| Page 1 Line 1 | "Total ozone column measurements can be made using""                |
|---------------|---------------------------------------------------------------------|
| Page 1 Line 2 | "In 2003,"                                                          |
| Page 1 Line 3 | " and since 2011 it has transferred"                                |
| Page 1 Line 4 | " calibration, mainly to other European Brewers, using Brewer #185" |
| Page 1 Line 5 | " annually, alternating"                                            |
| Page 1 Line 6 | " focused on reporting on the stability "                           |

| Page 1 Line 10 | " to the data from the Izana triad allows the comparison of the"                                                                                                                                                                                                                                                                                                                                                                                                                                                                                                                                                                                                                                                        |
|----------------|-------------------------------------------------------------------------------------------------------------------------------------------------------------------------------------------------------------------------------------------------------------------------------------------------------------------------------------------------------------------------------------------------------------------------------------------------------------------------------------------------------------------------------------------------------------------------------------------------------------------------------------------------------------------------------------------------------------------------|
| Page 1 Line 13 | " standard deviation of the mean equal"
"In opposite, using" Suggest " Alternatively, using the procedure used
to analyze the data from the Arosa Triad"                                                                                                                                                                                                                                                                                                                                                                                                                                                                                                                                                          |
| Page 1 Line 14 | " presents a relative standard deviation of about"                                                                                                                                                                                                                                                                                                                                                                                                                                                                                                                                                                                                                                                                      |
| Page 1 Line 15 | " method used for the data from the World Triad Reference"                                                                                                                                                                                                                                                                                                                                                                                                                                                                                                                                                                                                                                                              |
| Page 1 Line 16 | " gives monthly values of 0.3%"                                                                                                                                                                                                                                                                                                                                                                                                                                                                                                                                                                                                                                                                                         |
| Page 1 Line 17 | " datasets were analyzed" " second only included "                                                                                                                                                                                                                                                                                                                                                                                                                                                                                                                                                                                                                                                                      |
| Page 1 Line 18 | "Furthermore, this paper also describes the Langley method used to
determine the Extra-Terrestrial Constants (ETC) for the RBCC-E Triad,
the necessary first step toward accurate ozone measurement."                                                                                                                                                                                                                                                                                                                                                                                                                                                                                                             |
| Page 2 Line 2  | "Until a few decades ago, it was thought that the ozone concentration was constant in the stratosphere. However, after the discovery of the hole in the ozone layer in the mid-1980s, this idea was discarded (Farman et al., 1985)". This is a problematic statement. Ozone is variable, it has annual cycles - and longer-term cycles - and a non-uniform global distribution. It is not clear how to accurately express the simple thought the authors are searching for. Perhaps something like "Historical measurement s - pre 1980 - indicated that the morphology of ozone was not changing significantly with time. However, the Antarctic measurements of Farman et al., published in 1985, changed that view" |
| Page 2 Line 4  | "Concerns related to the negative effects that UV radiation can have on terrestrial life"                                                                                                                                                                                                                                                                                                                                                                                                                                                                                                                                                                                                                               |
| Page 2 Line 6  | " agents that led to this decrease"                                                                                                                                                                                                                                                                                                                                                                                                                                                                                                                                                                                                                                                                                     |
| Page 2 Line 8  | " total ozone column abundance"                                                                                                                                                                                                                                                                                                                                                                                                                                                                                                                                                                                                                                                                                         |
| Page 2 Line 9  | " Fioletov et al., 2005." And reference: Fioletov, V. E.; J. B. Kerr, C.T. McElroy, D.I. Wardle, V. Savastiouk, and T.S. Grajnar, The Brewer reference triad, Geophys. Res. Lett., 32, 10.1029, 2005.                                                                                                                                                                                                                                                                                                                                                                                                                                                                                                                   |
| Page 2 Line 12 | Fioletov, V. E.: Comparison of Brewer ultraviolet irradiance
measurements with total ozone mapping spectrometer satellite retrievals,
Optical Engineering, 41, 3051, https://doi.org/10.1117/1.1516818,
http://opticalengineering.spiedigitallibrary.org/article.aspx?doi=10.1117/1.
1516818, 2002. This is not a good reference for Brewer aerosol                                                                                                                                                                                                                                                                                                                                                         |

|                | measurements. It is, however, appropriate for UV Irradiance
measurements. Suggest that the references be segregated and repeatd if
necessary to make this clear. |
|----------------|------------------------------------------------------------------------------------------------------------------------------------------------------------------------|
| Page 2 Line 14 | " wavelenght separation" Spelling: " wavelength separation"
" spectral bands [regions?]" " These bands"                                                             |
| Page 2 Line 16 | "Although the prototype Brewer was developed in the early 1970s"
" has had on-going technical improvements to improve its accuracy"                                 |
| Page 2 Line 20 | Delete "concentration"                                                                                                                                                 |
| Page 2 Line 21 | Capitalize "Brewers present this"                                                                                                                                      |
| Page 2 Line 24 | " consisting of"                                                                                                                                                       |
| Page 2 Line 26 | " (Fioletov et al., 2005)"                                                                                                                                             |
| Page 2 Line 27 | " parallel with the World Triad Reference"                                                                                                                             |
| Page 2 Line 30 | " are calibrated by comparison with the travelling"                                                                                                                    |
| Page 3 Line 9  | " Observatory is located on the island of Tenerife"                                                                                                                    |
| Page 3 Line 15 | " campaigns through the travelling "                                                                                                                                   |
| Page 3 Line 16 | "These values were calculated using measurements in a range where
Brewer #017 measurements are not strongly affected by stray light"                                |
| Page 3 Line 23 | " each Brewer at its local station"                                                                                                                                    |
| Page 3 Line 32 | " which will allow the calculation of the TOC in near real time"                                                                                                       |
| Page 4 Line 1  | "Currently, approximately 40 Brewers"                                                                                                                                  |
| Page 4 Line 9  | "Also in this section the results of a study on the behavior of the RBCC-E
Triad as a function of SZA range at which the measurements were
performed."           |
| Page 4 Line 13 | " four spectral bands which"                                                                                                                                           |
| Page 4 Line 14 | " with local maximum and minimum ozone absorption cross-sections."
"The light intensity"                                                                            |
| Page 4 Line 16 | " the contribution of Rayleigh"                                                                                                                                        |

| Page 4 Line 17 | " more than one band is" This is not the main reason. The ratio technique eliminates the dependence of the absorption function on absolute intensity signal and the right wavelength weighting stops aerosol from interfering." 17 and 18 need a little more work. A fifth wavelength is included to measure $SO_2$                                                                                     |
|----------------|---------------------------------------------------------------------------------------------------------------------------------------------------------------------------------------------------------------------------------------------------------------------------------------------------------------------------------------------------------------------------------------------------------|
| Page 4 Line 19 | Suggest: "The Beer-Lambert law"                                                                                                                                                                                                                                                                                                                                                                         |
| Page 4 Line 22 | "where $\tau$ is the gas concentration" 1. $\tau$ is used in the literature for
optical depth, 2. X is normally used for ozone amount, not in
concentration (molecules/cm 3 ) but in cm (or other length units) to go along
with $\alpha$ in cm -1 . The airmass for ozone is usually written as $\mu$ to
distinguish it from the airmass for Rayleigh scattering, m. |
| Page 4 Line 29 | The equations in the PDF did not fare well.                                                                                                                                                                                                                                                                                                                                                             |
| Page 4 Line 15 | Measurements on slits 2 to 6 are used for $SO_2$ and ozone.                                                                                                                                                                                                                                                                                                                                             |
| Page 4 Line 24 | See earlier notes on variable names.                                                                                                                                                                                                                                                                                                                                                                    |

---

## Author Comment (AC1) · 14 Apr 2018

We would like to thank Referee1 for all his constructive suggestions and comments. They have been quite useful to improve the paper. We include as additional information in a zip file which contains:

1) Answer-Referee1: Pdf file with the answers to your questions/suggestions.( We also show them here).

2) Stability-diff: Pdf file where all changes are highlighted. Note that it also contains the changes of the other referees.

[Figure]

3) Manuscript-final: Pdf file with all changes introduced (not highlighted)

4) Figures have been modificated in the manuscript, we incluided its last version (.eps).

**General Comments**

**1   Figures**

In all the figures a colour legend has been introduced. For this reason, these are slightly different to the previous version of the article submitted.

**2   Section 2 "Theoretical approach"**

All referee have indicated that the section 2, where an approach to ozone retrieval (DS routine, slits) and Langley-technique are introduced, is confused. Therefore, it has been rewritten taking into account all the suggestions. Also, a new figure has been introduced.

**3   Grammatical errors**

The authors appreciate the grammatical corrections indicated which have been introduced in the text. A co-worker, who is a native English speaker, has helped us with the use of language. In the new version, some paragraphs and sentences were modified to get a more fluid text .

We have used both "behavior" and "behaviour". In order to have a more consistent article we have replaced the word "behavior" by "behaviour" in the text.

-Similarly, "centre" by "center". - P1, l9: add ...from "two different" methods previously...

**done**

- P1, l10: it should already be mentioned here that the World Reference is the Toronto triad. **done**

- P2, l5: "on" instead "in". **done**

- P2, l6: "cause" instead "produce". **done**

- P2, l9 and in references P20, l4: "Varotsos" instead "varotsos". **done**

- P2, l9: ". . .. . .which are considered to be reference instruments. . .." Is not correct. The triads or Brewer #17 are references (or standards). May be "basic instruments in the global network" or similar is better. **done**

- P2, l10: "Brewer spectrophotometers are widely used" since when? **This sentence was rewritten.**

- P2, l12/13: "ozone concentration" is not correct, "TOC" is better. **done**

- P2, l16: Why is "although" used? Better "After the development of the first Brewer in the early 1980s it has had continuous..."**This sentence was rewritten.**

- P2, l21: 70 degrees as limit for single Brewer observations seems small, as it corre-spondsto $\mu$-values of smaller than 3. Even single Brewers can measure reliable TOC up to 75 degrees ($\mu$-values around 3.5) under normally clear air conditions. Perhaps it would be good, to describe the different stray light issues: internal straylight-problem, which is larger for single Brewers, and external stray light (diffuse sky light around the sun), which is similar for all Brewers, but larger for Dobsons. The effect is the same in both cases: drop of TOC, when the SZA ($\mu$) increases and gets large depending on instrument and turbidity. **This sentence was rewritten. We think that is better to**

**say that ".. decrease in the TOC measurement at large ozone slant column"**

- P2, l24: The calibration of the Brewer "in the first years" is traceable.

- P2, l31: by the "manufacturer" Kipp & Zonen.**done**

- P3, l5: Add "triad" behind RBCC-E.**done**

- P3, l9: Replace "in the island" with "on the island".**done**

- P3, l16: "These values have been".**done**

- P3, l20: "more than 150 Brewers have been calibrated". **done**

- P4, l13 ff, p5: see under general comments (SO2!). The reference Dobson 1957 is out of place here and I think anywhere in the text. So omit it in the references. The presentation of the corresponding absorption spectra (O3 and SO2) might be helpful.**This section was rewritten**

- P5, l15: DS observations are done in 6, not in 7 slits: 4 ozone, 1 SO2 and 1 dark, the seventh slit is for HG-test. **This section was rewritten**

- P4/5/6: Use of m instead of $\mu$ is not correct. **done**

- P6, l7: "clear" instead of "clean"-sky.**done**

- P6, l11: Why does the use of 1/m (should also be $\mu$) allow obtaining two ETCs? I thought it is the splitting in fore- and afternoon observations.This section was rewritten

- P6, l 32-35: no complete sentence. **The sentence was corrected**

- P8, l6: Giving a number for the low standard deviation would be helpful.**done** - P8, l9: "Langley technique can be used" instead of "Langley-technique is used".**done**

- P8, l18: correct "a an".**done**

Chapter 4: - P10, 4: "mean of daily differences" or "daily mean of the difference"? **This sentence was rewritten**

- P10,l5: Replace "analyzed" with "analyze".**done**

- P10, l 10; replace "inconvenient" with "inconvenience".**done**

- P10, l11 and P11, l11: isn't Izana a subtropical station? **This sentences were rewritten**

- P10, l15: "another" instead of "other".**done**

- P11, figure 4: The "A" from Eq. 10 is missed. Shouldn't it be placed behind "ozone reference value"? The colors are not assigned to the Brewers.**done**

- P11, last section, P12, first section: Sorry, but I have problems to understand what is meant. The beginning of two consecutive sentence with "Therefore" does not sound good. It is described that the polynomial method show similar results regardless of the data set and the order of the fit. Why is then the daily mean method more appropriate? **This paragraph was rewritten**

- P15, Figure 6: The order of the panels does not coincide with the order of the mentioned methods in the figure caption. **done**

- P15, l10: correct "surprissing". **done**

**4   Answers at your specific questions:**

**The description of the triad system is not profound and clear. Where is the second triad located? In Toronto too? How are this triad and in addition the travelling Brewer No. 17 calibrated? It is good to know how stable the RBCC-E triad is, but what about its accuracy? The agreement between this triad and the World Reference Triad is confirmed by the 0.5% - agreement with #017. But how good is the agreement of #017 with World Reference Triad? When was its last absolute Langley calibration performed?** In the introduction we made reference to that

both (World Reference and the Second Triads) work in parallel in Toronto. In order to clarify this questions, a new reference has been included in the text. An oral presentation where is shown some graphic about the agreement between both triads, and also some photos of the Brewers have been added too. In addiction, V. Savastiouk, who operated the Brewer #017, have indicated that"Brewer #017 is regularly compared with the World triad Reference in Toronto it is also absolutely and independently calibrated at the Mauna Loa Observatory in Hawaii, US. Some history of such calibrations can be found at http://www.io3.ca/Calibrations/Brewer/017".(see interactive discussion of our article.)

**P2, l21: 70 degrees as limit for single Brewer observations seems small, as it corresponds to $\mu$-values of smaller than 3. Even single Brewers can measure reliable TOC up to 75 degrees ($\mu$-values around 3.5) under normally clear air conditions. Perhaps it would be good, to describe the different stray light issues: internal straylight-problem, which is larger for single Brewers, and external stray light (diffuse sky light around the sun), which is similar for all Brewers, but larger for Dobsons. The effect is the same in both cases: drop of TOC, when the SZA ($\mu$) increases and gets large depending on instrument and turbidity.** Yes, perhaps this value is a very extreme lower limit. This sentence was rewritten. We think that is better to say that "...decrease in the TOC measurement at large ozone slant column".

**- P8, l6: Giving a number for the low standard deviation would be helpful.** Following the suggestion of other referee this sentence was rewritten "despite this annual behaviour, the ozone presents a lower daily variability as indicated our measurements".

**- P8, l17: The criterion lower than 0.6 under item 4 is not clear: standard deviation of 0.6 DU? Seems to be a very low standard deviation for a day with small ozone variation.** Yes, this condition was introduced in the article by mistake. It was written in a previous version (internal draft) but it was not deleted from the uploaded article.

**- P8, l20: what does "(condition 3 above)" mean in the context with simultane-**

**ous? Does it mean, that in addition to condition 3, these selected measurements should be simultaneous?.** rephrased as additional condition.

**- P10, l8/9: what does slight mechanical miscalculation mean?** Sorry, maybe this sentence is not the best translation into English. The text has been modified as follows:

Old version: "...it should be noted that the presence of slight mechanical miscalculations in the instrument...."

New Version: "... it should be noted that the presence of small drifts by its continued operation of the instrument...."

**- P11, figure 4: The "A" from Eq. 10 is missed. Shouldn't it be placed behind "ozone reference value"? The colors are not assigned to the Brewers** A new figure with a colour legend has been upload. The "A" was included in the figure caption.

**- P11, last section, P12, first section: Sorry, but I have problems to understand what is meant. The beginning of two consecutive sentence with "Therefore" does not sound good. It is described that the polynomial method show similar results regardless of the data set and the order of the fit. Why is then the daily mean method more appropriate?** This paragraph was rewritten because, as you indicate, it is confusing. The "therefore" were replaced by other connecters.

**P13, Fig.5: Again the colors are not assigned to the Brewers** A new figure with a colour legend have been upload.

**- P12 – P15: What about Brewer #183? Tables 3 and 4 (Data set 1) and Figures 5 and 6 show a larger ratio in 2007, which cannot be seen in the graph of Arosa method. Only the table 4 gives a negative shift median and larger percentile numbers for Brewer #183 data set 1. Is the reason known? The same is valid for the larger scattering of all RBCC-E Brewers around 2010.** Thank you! If the data in this graph was wrong. We have reviewed our calculations and now the results are more coherent with the previous graphs.

**2010: issues on 183, during that time only 157 and 185 were used for reference.** In 2010, the brewer #183 had some problem with their micrometer during some months. As in the previous questions, the fit can hide this problem. We have introduced this reference where some plot about this are shown. *Roozendael, M. V., Köhler, U., Pappalardo, G., Kyrö, E., Redondas, A., Wittrock, F., Amodeo, A. and Pinardi, G.: CEOS Intercalibration of Ground-Based Spectrometers and Lidars: Final report. http://repositorio.aemet.es/handle/20.500.11765/8886 (Accessed 5 April 2018), 2013.*

**- P14, l9 (and in abstract too): Where do the numbers for the RBCC-E triad come from? I cannot find them in the tables.** The values are the serial numbers of the brewer

**- P14, l10: how is 40% lower dispersion determined?** This value was calculated as the ratio between both relative deviation (0.47-0.27)/0.47=0.42

**- P14, l12: Is the 0.3% value for RBCC-E also calculated in the 2004 – 2012 period of the Stübi-paper** We're sorry, but we cannot calculate that data. As we indicated in the introduction, brewers # 183 and # 185 were installed in 2005, hence there is not available data in the year 2004. In the article, the value reported corresponds with the period 2005-2016. In addition, it can be confusing to give values that do not correspond with the years defined in the datasets.

**- P15, Figure 6: The order of the panels does not coincide with the order of the mentioned methods in the figure caption.** We have rewritten the Figure caption in the correct order.

**- P15/16: The explanation for the larger scattering at high sun is good and comprehensible. But the obvious difference of low sun in the forenoon and in the afternoon is neither mentioned nor explained. Is there also an explanation, why Brewer #183 shows larger scattering in the relative difference than the other two Brewers?** Honestly, we think that the Brewer \#183 presents this large scattering because it was damaged in 2007, during the "Delta" storm and during 2008 it had an

irregular behaviour and this data was used in this analysis. In addiction, this differ-ence is minimal or negligible and does not affect the calibration of the Triad. Moreover, this result is consistent with those presented in tables 3 and 4 where it is found that brewer 183 has the poorer values. It seems logical to think that if in average values (daily or monthly) this brewer is the worst, also depending on the SZA this difference is reflected.

**- P16/17: The values in table 5 are not clear. Where do they come from? Fioletov daily of 0.47% is mentioned on P16, l8. In the same context the RBCC-E triad number is given as 0.41% in the Conclusion, but as 0.37 in the table, which is confusing.**

In order to clarify this question, a new table was generated.

Please also note the supplement to this comment:
https://www.atmos-meas-tech-discuss.net/amt-2017-460/amt-2017-460-AC1-supplement.zip

---

## Author Comment (AC3) · 14 Apr 2018

We would like to thank V.Savastiouk for all his constructive suggestions and comments. They have been quite useful to improve the paper. We include as additional information in a zip file which contains:

1) Answer-V.Savastiouk: Pdf file with the answers to your questions/suggestions.( We also show them here).

2) Stability-diff: Pdf file where all changes are highlighted. Note that it also contains the changes of the other referees.

[Figure]

3) Manuscript-final: Pdf file with all changes introduced (not highlighted)

4) Figures have been modificated in the manuscript, we incluided its last version (.eps).

**1 General corrections**

**2 Figures**

In all the figures a colour legend has been introduced. For this reason, these are slightly different to the previous version of the article submitted.

**3 Grammatical corrections**

The authors appreciate the grammatical corrections indicated which have been introduced in the text. A co-worker, who is a native English speaker, has helped us with the use of language. In the new version, some paragraphs and sentences were modified to get a more fluid text.

- We have used both "behavior" and "behaviour" words. In order to have a more consistent article we have replaced "behavior" by "behaviour" in the text. Similarly, "centre" by "center".

- In addition, the expression "ozone concentration" was replaced by "Total ozone column (TOC)".

- The word "lines" was replaced by "bands" to indicate the wavelengths where the brewer measured.
**4 References**

New references have been included in the text where you and the other referees have indicated. However, we think that introducing a reference in the abstract as you suggest in your report is not habitual.

**5 Abstract**

**P1- L9**. This sentence was rewritten taking into account your suggestions. Also, the standard deviation symbol $\sigma$ has been introduced together with the values reported.

**6 Introduction**

**P2- L5**. The sentence where we suggest that "... several countries agreed to reduce the agents that produces the decrease of ..." was modified and, the number of countries agreed (197) was included.

**P2- L12**. The expression "Brewer is a spectrophotometer" was replaced by "Brewer ozone spectrophotometer". Also, this paragraph were rewritten.

**P2- L21**. 70 degrees as limit for single Brewer observations seems small, as it corresponds to $\mu$-values of smaller than 3. We think that is better to say that " The single Brewer presents this problem for large ozone slant column (OSC)".

**P2- L32**. We think that is correct to duplicate the sentence "Since November 2003 and within the World Meteorological Organization (WMO) and the Global Atmosphere Watch ...." because in this paragraph it is used to talk about the history of the RBCC-E and, obviously, you must say: when was created it?, when the brewers were installed

?, which is our travelling Brewer?, etc.

**P3- L11** We say that Izaña has "clean air and clear sky conditions around all the year and offers excellent conditions to perform the Langley-technique". As you indicate we have not taken into account the dust intrusions. Therefore, the sentence was rewritten: "This ensures clean air and clear sky conditions around all the year and offers excellent conditions to perform the Langley technique, except for some days where the Saharan dust intrusions make difficult to measure the direct solar radiation".

**P3- L12**. The sentence "Moreover, comparisons with the World Triad reference are carried out regularly..." was rewritten as " Moreover, the traceability between the RBCC-E Triad and the World Triad Reference is checked during the calibration campaigns through the travelling references #185 and #017."

**P3- L19**. The expression "annually alternating" was replaced by "annually, alternating"

**P3- L23**. The sentence "... on the measurements performed by each Brewer in its local station" was rewritten.

**P4- L4**. The sentence "... the precision between the measurements" was rewritten as "how similar are the measurements made by the Brewers "

**7  Theoretical Approach**

All referees have indicated that the section 2 is confused. Therefore, it has been rewritten taking into account all the suggestions about the approach to ozone retrieval (DS routine, slits) and Langley-technique. Also, a new figure has been introduced and the equations were modified, adding the symbol $\mu$. In addition, a clearer explanation about the procedure of the Langley technique was written. In it, we relate the Langley with the inverse of $\mu$, and also explain how it affects the aerosols in this technique.

Now, we hope that the order of the section will be better.

**8 Ozone and dataset selected**

**P8- L6**. The sentence "Despite this annual behaviour, the ozone is stable during the day, with a low standard deviation for the recorded data." was rewritten as "Despite this annual behaviour, the ozone presents a lower daily variability as indicated by our measurements".

**P8- L17**. The criterion lower than 0.6 under item 4 is not clear: standard deviation of 0.6 DU? Seems to be a very low standard deviation for a day with small ozone variation.

Yes, this condition was introduced in the article by mistake. It was written in a previous version (internal draft) but it was not deleted from the uploaded article.

**9 Results and discussion**

**P10- L8/9. what does slight mechanical miscalculation mean?**

Sorry, maybe this sentence is not the best translation into English. The text has been modified as follows:

Old version: "... it should be noted that the presence of slight mechanical miscalculations in the instrument. . . ."

New Version: "... it should be noted that the presence of small drifts by its continued operation of the instrument. . . ."

**P12 -L12**. The equation "mean value" was correct.

**P10- L15**. The expression "our experience suggest" was rewritten as "our experience

suggests".

**10 Conclusions**

Some sentences were rewritten. The standard deviation symbol $\sigma$ has been introduced to identify the values reported and a new table where all values are summarised is also included.

Please also note the supplement to this comment:
https://www.atmos-meas-tech-discuss.net/amt-2017-460/amt-2017-460-AC3-supplement.zip

---

## Author Comment (AC5) · 14 Apr 2018

Dear Referee 1:

**The description of the triad system is not profound and clear. Where is the second triad located? In Toronto too? How are this triad and in addition the travelling Brewer No. 17 calibrated? It is good to know how stable the RBCC-E triad is, but what about its accuracy? The agreement between this triad and the World Reference Triad is confirmed by the 0.5% - agreement with #017. But how good is the agreement of #017 with World Reference Triad? When was its last absolute Langley calibration performed?**

In the introduction we made reference to that both (World Reference and the Second Triads) work in parallel in Toronto. In order to clarify this questions, a new reference has been included in the text. An oral presentation where is shown some graphic about the agreement between both triads, and also some photos of the Brewers have been added too.

We think that V. Savastiouk, who operated the Brewer #017, is the best person to answer about the agreement between the World Triad Reference and this Brewer.

"V. Savastiouk have indicated that"Brewer #017 is regularly compared with the World triad Reference in Toronto it is also absolutely and independently calibrated at the Mauna Loa Observatory in Hawaii, US. Some history of such calibrations can be found at http://www.io3.ca/Calibrations/Brewer/017".(see his interactive answer)

---

## Author Response (AR1)

**Stability of the RBCC-E Triad during the period 2005 – 2016**

Sergio F. León-Luis[1,2], Alberto Redondas[1,2], Virgilio Carreño[1,2], Javier López-Solano[2,3], Alberto Berjón[2,3], Bentorey Hernández-Cruz[1,3], and Daniel Santana-Díaz[2,3]

[1]Izaña Atmospheric Research Center, Agencia Estatal de Meteorología, Tenerife, Spain
[2]Regional Brewer Calibration Center for Europe, Izaña Atmospheric Research Center, Tenerife, Spain
[3]Departamento de Ingeniería Industrial, Universidad de La Laguna, Tenerife, Spain

*Correspondence to:* Alberto Redondas (aredondasm@aemet.es)

**Answer to Tom McElroy**

Thank you for your review. According to your suggestions and those of other referees, we submit a revised version of our manuscript entitled "Stability of the Regional Calibration Center for Europe Triad during the period 2005-2016" by Sergio Fabián León-Luis, Alberto Redondas, Virgilio Carreño, Javier López-Solano, Alberto Berjón, Bentorey Hernández-Cruz and Daniel Santana-Díaz. All issues brought up by the reviewers have been addressed.

**Figures**

In all the figures a colour legend has been introduced. For this reason, these are slightly different to the previous version of the article submitted.

**Section 2 "Theoretical approach"**

All referees have indicated that the section 2 is confused. Therefore, it has been rewritten taking into account all suggestions indicated by the referees. Also, a new figure has been introduced.

**Grammatical errors**

The authors appreciate the grammatical corrections indicated which have been introduced in the text. A co-worker, who is a native English speaker, has helped us with the use of language. In the new version, some paragraphs and sentences were modified to get a more fluid text.

We have were used both "behavior" and "behaviour". In order to have a more consistent article we have replaced the word "behavior" by "behaviour" in the text.

-Similarly, "tractability" by "traceability". -Similarly, "centre" by "center".

Page 1 Line 1 "Total ozone column measurements can be made using ..." **The text was modified**

Page 1 Line 2 "In 2003, ..." **The text was modified**

Page 1 Line 3 "... and since 2011 it has transferred ..." **The text was modified**

Page 1 Line 4 "... calibration, mainly to other European Brewers, using Brewer #185 ..."**The text was modified**

Page 1 Line 5 "... annually, alternating ..."**The text was modified**

Page 1 Line 6 "... focused on reporting on the stability... "**The text was modified**

Page 1 Line 10 "... to the data from the Izana triad allows the comparison of the ..." **The text was modified**

Page 1 Line 13 "... standard deviation of the mean equal ..." **The text was modified**

"In opposite, using..." Suggest "... Alternatively, using the procedure used to analyze the data from the Arosa Triad ..." **The text was modified**

Page 1 Line 14 "... presents a relative standard deviation of about ..." **The text was modified**

Page 1 Line 15 "... method used for the data from the World Triad Reference ..." **The text was modified**

Page 1 Line 16 "... gives monthly values of 0.3% ..." **The text was modified**

Page 1 Line 17 "... datasets were analyzed ..." "... second only included... "**The text was modified**

Page 1 Line 18 "Furthermore, this paper also describes the Langley method used to determine the Extra-Terrestrial Constants (ETC) for the RBCC-E Triad, the necessary first step toward accurate ozone measurement."**The text was modified**

Page 2 Line 2 "Until a few decades ago, it was thought that the ozone concentration was constant in the stratosphere. However, after the discovery of the hole in the ozone layer in the mid-1980s, this idea was discarded (Farman et al., 1985)". This is a problematic statement. Ozone is variable, it has annual cycles - and longer-term cycles - and a non-uniform global distribution. It is not clear how to accurately express the simple thought the authors are searching for. Perhaps something like "Historical measurement s - pre 1980 - indicated that the morphology of ozone was not changing significantly with time. However, the Antarctic measurements of Farman et al., published in 1985, changed that view...." **The text was modified**

Page 2 Line 4 "Concerns related to the negative effects that UV radiation can have on terrestrial life ..."**The text was modified**

Page 2 Line 6 "... agents that led to this decrease ..." **The text was modified**

Page 2 Line 8 ".. total ozone column abundance ..." **The text was modified**

Page 2 Line 9 "... Fioletov et al., 2005." And reference: Fioletov, V. E.; J. B. Kerr, C.T. McElroy, D.I. Wardle, V. Savastiouk, and T.S. Grajnar, The Brewer reference triad, Geophys. Res. Lett., 32, 10.1029, 2005. **The reference was modified**

Page 2 Line 12 Fioletov, V. E.: Comparison of Brewer ultraviolet irradiance measurements with total ozone mapping spectrometer satellite retrievals, Optical Engineering, 41, 3051, https://doi.org/10.1117/1.1516818, http://opticalengineering.spiedigitallibrary.o This is not a good reference for Brewer aerosol measurements. It is, however, appropriate for UV Irradiance measurements. Suggest that the references be segregated and repeated ifnecessary to make this clear. **The reference was relocated**

Page 2 Line 14 "... wavelenght separation... " Spelling: "... wavelength separation... " **The text was modified**

Page 2 Line 16 "Although the prototype Brewer was developed in the early 1970s ..." "... has had on-going technical improvements to improve its accuracy..." **The text was modified**

Page 2 Line 20 Delete "concentration" **The text was modified**

Page 2 Line 21 Capitalize "Brewers present this ..." **The text was modified**

Page 2 Line 24 "... consisting of. ..." **The text was modified**

Page 2 Line 26 "... (Fioletov et al., 2005) ..." **The text was modified**

Page 2 Line 27 "... parallel with the World Triad Reference ..." **The text was modified**

Page 2 Line 30 "... are calibrated by comparison with the travelling ..." **The text was modified**

Page 3 Line 9 "... Observatory is located on the island of Tenerife ..." **The text was modified**

Page 3 Line 15 "... campaigns through the travelling... " **The text was modified**

Page 3 Line 16 "These values were calculated using measurements in a range where Brewer #017 measurements are not strongly affected by stray light ..." **The text was modified**

Page 3 Line 23 "... each Brewer at its local station ..." **The text was modified**

Page 3 Line 32 "... which will allow the calculation of the TOC in near real time ..." **The text was modified**

Page 4 Line 1 "Currently, approximately 40 Brewers ..." **The text was modified**

Page 4 Line 9 "Also in this section the results of a study on the behavior of the RBCC-E Triad as a function of SZA range at which the measurements were performed." **The text was modified**

Page 4 Line 13 "... four spectral bands which ..." **The text was modified**

Page 4 Line 14 "... with local maximum and minimum ozone absorption cross-sections." **The text was modified**

Page 4 Line 16 "... the contribution of Rayleigh ... **The text was modified**

Page 4 Line 17 "... more than one band is ..." This is not the main reason. The ratio technique eliminates the dependence of the absorption function on absolute intensity signal and the right wavelength weighting stops aerosol from interfering." 17 and 18 need a little more work. A fifth wavelength is included to measure SO2 **This section was modified**

Page 4 Line 19 Suggest: "The Beer-Lambert law ..." **The text was modified**

Page 4 Line 22 "where J is the gas concentration ..." 1. J is used in the literature for optical depth, 2. X is normally used for ozone amount, not in concentration (molecules/cm3 ) but in cm (or other length units) to go along with " in cm-1. The airmass for ozone is usually written as : to distinguish it from the airmass for Rayleigh scattering, m. **The section 2 was modified and your suggestions were added**

Page 4 Line 29 The equations in the PDF did not fare well. **The section 2 was modified and your suggestions were added**

Page 4 Line 15 Measurements on slits 2 to 6 are used for SO2 and ozone. **The section 2 was modified and your suggestions were added**

Page 4 Line 24 See earlier notes on variable names. **The section 2 was modified and your suggestions were added**

On behalf of the co-authors, thank you very much for your suggestions.

Alberto Redondas Marrero.

Regional Brewer calibration Center.

Izaña Atmospheric Research Center

AEMET- Meteorological State Agency, Spain.

**Stability of the RBCC-E Triad during the period 2005 – 2016**

Sergio F. León-Luis[1,2], Alberto Redondas[1,2], Virgilio Carreño[1,2], Javier López-Solano[2,3], Alberto Berjón[2,3], Bentorey Hernández-Cruz[1,3], and Daniel Santana-Díaz[2,3]

[1]Izaña Atmospheric Research Center, Agencia Estatal de Meteorología, Tenerife, Spain
[2]Regional Brewer Calibration Center for Europe, Izaña Atmospheric Research Center, Tenerife, Spain
[3]Departamento de Ingeniería Industrial, Universidad de La Laguna, Tenerife, Spain

*Correspondence to:* Alberto Redondas (aredondasm@aemet.es)

**Answer to Referee #1**

Thank you for your review. According to your suggestions and those of other referees, we submit a revised version of our manuscript entitled "Stability of the Regional Calibration Center for Europe Triad during the period 2005-2016" by Sergio Fabián León-Luis, Alberto Redondas, Virgilio Carreño, Javier López-Solano, Alberto Berjón, Bentorey Hernández-Cruz and
5 Daniel Santana-Díaz. All issues brought up by the reviewers have been addressed.

**Figures**

In all the figures a colour legend has been introduced. For this reason, these are slightly different to the previous version of the article submitted.

**Section 2 "Theoretical approach"**

10 All referee have indicated that the section 2, where an approach to ozone retrieval (DS routine, slits) and Langley-technique are introduced, is confused. Therefore, it has been rewritten taking into account all the suggestions. Also, a new figure has been introduced.

**Grammatical errors**

The authors appreciate the grammatical corrections indicated which have been introduced in the text. A co-worker, who is
15 a native English speaker, has helped us with the use of language. In the new version, some paragraphs and sentences were modified to get a more fluid text .

We have used both "behavior" and "behaviour". In order to have a more consistent article we have replaced the word "behavior" by "behaviour" in the text.

-Similarly, "centre" by "center". - P1, l9: add ...from "two different" methods previously... **done**

- P1, l10: it should already be mentioned here that the World Reference is the Toronto triad. **done**

- P2, l5: "on" instead "in". **done**

- P2, l6: "cause" instead "produce". **done**

- P2, l9 and in references P20, l4: "Varotsos" instead "varotsos". **done**

- P2, l9: ". . .. . .which are considered to be reference instruments. . .." Is not correct. The triads or Brewer #17 are references (or standards). May be "basic instruments in the global network" or similar is better. **done**

- P2, l10: "Brewer spectrophotometers are widely used" since when? **This sentence was rewritten.**

- P2, l12/13: "ozone concentration" is not correct, "TOC" is better. **done**

- P2, l16: Why is "although" used? Better "After the development of the first Brewer in the early 1980s it has had continuous..."**This sentence was rewritten.**

- P2, l21: 70 degrees as limit for single Brewer observations seems small, as it correspondsto µ-values of smaller than 3. Even single Brewers can measure reliable TOC up to 75 degrees (µ-values around 3.5) under normally clear air conditions. Perhaps it would be good, to describe the different stray light issues: internal straylight-problem, which is larger for single Brewers, and external stray light (diffuse sky light around the sun), which is similar for all Brewers, but larger for Dobsons. The effect is the same in both cases: drop of TOC, when the SZA (µ) increases and gets large depending on instrument and turbidity. **This sentence was rewritten. We think that is better to say that ".. decrease in the TOC measurement at large ozone slant column"**

- P2, l24: The calibration of the Brewer "in the first years" is traceable.

- P2, l31: by the "manufacturer" Kipp & Zonen.**done**

- P3, l5: Add "triad" behind RBCC-E.**done**

- P3, l9: Replace "in the island" with "on the island".**done**

- P3, l16: "These values have been".**done**

- P3, l20: "more than 150 Brewers have been calibrated". **done**

- P4, l13 ff, p5: see under general comments (SO2!). The reference Dobson 1957 is out of place here and I think anywhere in the text. So omit it in the references. The presentation of the corresponding absorption spectra (O3 and SO2) might be helpful.**This section was rewritten**

- P5, l15: DS observations are done in 6, not in 7 slits: 4 ozone, 1 SO2 and 1 dark, the seventh slit is for HG-test. **This section was rewritten**

- P4/5/6: Use of m instead of µ is not correct. **done**

- P6, l7: "clear" instead of "clean"-sky.**done**

- P6, l11: Why does the use of 1/m (should also be µ) allow obtaining two ETCs? I thought it is the splitting in fore- and afternoon observations.This section was rewritten

- P6, l 32-35: no complete sentence. **The sentence was corrected**

- P8, l6: Giving a number for the low standard deviation would be helpful.**done** - P8, l9: "Langley technique can be used" instead of "Langley-technique is used".**done**

- P8, l18: correct "a an".**done**

Chapter 4: - P10, 4: "mean of daily differences" or "daily mean of the difference"? **This sentence was rewritten**

- P10,l5: Replace "analyzed" with "analyze".**done**

- P10, l 10; replace "inconvenient" with "inconvenience".**done**

- P10, l11 and P11, l11: isn't Izana a subtropical station? **This sentences were rewritten**

- P10, l15: "another" instead of "other".**done**

- P11, figure 4: The "A" from Eq. 10 is missed. Shouldn't it be placed behind "ozone reference value"? The colors are not assigned to the Brewers.**done**

- P11, last section, P12, first section: Sorry, but I have problems to understand what is meant. The beginning of two consecutive sentence with "Therefore" does not sound good. It is described that the polynomial method show similar results regardless of the data set and the order of the fit. Why is then the daily mean method more appropriate? **This paragraph was rewritten**

- P15, Figure 6: The order of the panels does not coincide with the order of the mentioned methods in the figure caption. **done**

- P15, l10: correct "surprissing". **done**

**Answers at your specific questions:**

Comment: **The description of the triad system is not profound and clear. Where is the second triad located? In Toronto too? How are this triad and in addition the travelling Brewer No. 17 calibrated? It is good to know how stable the RBCC-E triad is, but what about its accuracy? The agreement between this triad and the World Reference Triad is confirmed by the 0.5% - agreement with #017. But how good is the agreement of #017 with World Reference Triad? When was its last absolute Langley calibration performed?**

Answer:  In the introduction we made reference to that both (World Reference and the Second Triads) work in parallel in Toronto. In order to clarify this questions, a new reference has been included in the text. An oral presentation where is shown some graphic about the agreement between both triads, and also some photos of the Brewers have been added too. In addiction, V. Savastiouk, who operated the Brewer #017, have indicated that"Brewer #017 is regularly compared with the World triad Reference in Toronto it is also absolutely and independently calibrated at the Mauna Loa Observatory in Hawaii, US. Some history of such calibrations can be found at http://www.io3.ca/Calibrations/Brewer/017".(see interactive discussion of our article.)

Comment: **P2, l21: 70 degrees as limit for single Brewer observations seems small, as it corresponds to µ-values of smaller than 3. Even single Brewers can measure reliable TOC up to 75 degrees (µ-values around 3.5) under normally clear air conditions. Perhaps it would be good, to describe the different stray light issues: internal straylight-**

**problem, which is larger for single Brewers, and external stray light (diffuse sky light around the sun), which is similar for all Brewers, but larger for Dobsons. The effect is the same in both cases: drop of TOC, when the SZA (μ) increases and gets large depending on instrument and turbidity.**

    **Answer:** Yes, perhaps this value is a very extreme lower limit. This sentence was rewritten. We think that is better to say that "...decrease in the TOC measurement at large ozone slant column".

**Comment:** **- P8, l6: Giving a number for the low standard deviation would be helpful.**

    **Answer:** Following the suggestion of other referee this sentence was rewritten "despite this annual behaviour, the ozone presents a lower daily variability as indicated our measurements".

**Comment:** **- P8, l17: The criterion lower than 0.6 under item 4 is not clear: standard deviation of 0.6 DU? Seems to be a very low standard deviation for a day with small ozone variation.**

    **Answer:** Yes, this condition was introduced in the article by mistake. It was written in a previous version (internal draft) but it was not deleted from the uploaded article.

**Comment:**

**- P8, l20: what does "(condition 3 above)" mean in the context with simultaneous? Does it mean, that in addition to condition 3, these selected measurements should be simultaneous?.**

    **Answer:** rephrased as additional condition.

**Comment:** **- P10, l8/9: what does slight mechanical miscalculation mean?**

    **Answer:** Sorry, maybe this sentence is not the best translation into English. The text has been modified as follows:

Old version: "...it should be noted that the presence of slight mechanical miscalculations in the instrument...."

New Version: "... it should be noted that the presence of small drifts by its continued operation of the instrument...."

**Comment:** **- P11, figure 4: The "A" from Eq. 10 is missed. Shouldn't it be placed behind "ozone reference value"? The colors are not assigned to the Brewers**

    **Answer:** A new figure with a colour legend has been upload. The "A" was included in the figure caption.

**Comment:** **- P11, last section, P12, first section: Sorry, but I have problems to understand what is meant. The beginning of two consecutive sentence with "Therefore" does not sound good. It is described that the polynomial method show similar results regardless of the data set and the order of the fit. Why is then the daily mean method more appropriate?**

    **Answer:** This paragraph was rewritten because, as you indicate, it is confusing. The "therefore" were replaced by other connecters.

**Comment: P13, Fig.5: Again the colors are not assigned to the Brewers**

   **Answer:** A new figure with a colour legend have been upload.

**Comment: - P12 – P15: What about Brewer #183? Tables 3 and 4 (Data set 1) and Figures 5 and 6 show a larger ratio in 2007, which cannot be seen in the graph of Arosa method. Only the table 4 gives a negative shift median and larger percentile numbers for Brewer #183 data set 1. Is the reason known? The same is valid for the larger scattering of all RBCC-E Brewers around 2010.**

   **Answer:** Thank you! If the data in this graph was wrong. We have reviewed our calculations and now the results are more coherent with the previous graphs.

**Comment:**

**2010: issues on 183, during that time only 157 and 185 were used for reference.**

   **Answer:** In 2010, the brewer #183 had some problem with their micrometer during some months. As in the previous questions, the fit can hide this problem. We have introduced this reference where some plot about this are shown. *Roozen-dael, M. V., Köhler, U., Pappalardo, G., Kyrö, E., Redondas, A., Wittrock, F., Amodeo, A. and Pinardi, G.: CEOS Intercal-ibration of Ground-Based Spectrometers and Lidars: Final report. http://repositorio.aemet.es/handle/20.500.11765/8886 (Accessed 5 April 2018), 2013.*

**Comment: - P14, l9 (and in abstract too): Where do the numbers for the RBCC-E triad come from? I cannot find them in the tables.**

   **Answer:** The values are the serial numbers of the brewer

**Comment: - P14, l10: how is 40% lower dispersion determined?**

   **Answer:** This value was calculated as the ratio between both relative deviation (0.47-0.27)/0.47=0.42

**Comment: - P14, l12: Is the 0.3% value for RBCC-E also calculated in the 2004 – 2012 period of the Stübi-paper**

   **Answer:** We're sorry, but we cannot calculate that data. As we indicated in the introduction, brewers # 183 and # 185 were installed in 2005, hence there is not available data in the year 2004. In the article, the value reported corresponds with the period 2005-2016. In addition, it can be confusing to give values that do not correspond with the years defined in the datasets.

**Comment: - P15, Figure 6: The order of the panels does not coincide with the order of the mentioned methods in the figure caption.**

   **Answer:** We have rewritten the Figure caption in the correct order.

**Comment: - P15/16: The explanation for the larger scattering at high sun is good and comprehensible. But the obvious difference of low sun in the forenoon and in the afternoon is neither mentioned nor explained. Is there also an**

**explanation, why Brewer #183 shows larger scattering in the relative difference than the other two Brewers?**

Answer: Honestly, we think that the Brewer \#183 presents this large scattering because it was damaged in 2007, during the "Delta" storm and during 2008 it had an irregular behaviour and this data was used in this analysis. In addiction, this difference is minimal or negligible and does not affect the calibration of the Triad. Moreover, this result is consistent with those presented in tables 3 and 4 where it is found that brewer 183 has the poorer values. It seems logical to think that if in average values (daily or monthly) this brewer is the worst, also depending on the SZA this difference is reflected.

Comment: **- P16/17: The values in table 5 are not clear. Where do they come from? Fioletov daily of 0.47% is mentioned on P16, l8. In the same context the RBCC-E triad number is given as 0.41% in the Conclusion, but as 0.37 in the table, which is confusing.**

Answer:

In order to clarify this question, a new table was generated.

On behalf of the co-authors, thank you very much for your suggestions.

Alberto Redondas Marrero.

Regional Brewer calibration Center.

Izaña Atmospheric Research Center

AEMET- Meteorological State Agency, Spain.

**Stability of the RBCC-E Triad during the period 2005 – 2016**

Sergio F. León-Luis[1,2], Alberto Redondas[1,2], Virgilio Carreño[1,2], Javier López-Solano[2,3], Alberto Berjón[2,3], Bentorey Hernández-Cruz[1,3], and Daniel Santana-Díaz[2,3]

[1]Izaña Atmospheric Research Center, Agencia Estatal de Meteorología, Tenerife, Spain
[2]Regional Brewer Calibration Center for Europe, Izaña Atmospheric Research Center, Tenerife, Spain
[3]Departamento de Ingeniería Industrial, Universidad de La Laguna, Tenerife, Spain

*Correspondence to:* Alberto Redondas (aredondasm@aemet.es)

**Answer to V. Savastiouk**

Thank you for your review. According to your suggestions and those of other referees, we submit a revised version of our manuscript entitled "Stability of the Regional Calibration Center for Europe Triad during the period 2005-2016" by Sergio Fabián León-Luis, Alberto Redondas, Virgilio Carreño, Javier López-Solano, Alberto Berjón, Bentorey Hernández-Cruz and Daniel Santana-Díaz. All issues brought up by the reviewers have been addressed.

**Figures**

In all the figures a colour legend has been introduced. For this reason, these are slightly different to the previous version of the article submitted.

**General corrections**

The authors appreciate the grammatical corrections indicated which have been introduced in the text. A co-worker, who is a native English speaker, has helped us with the use of language. In the new version, some paragraphs and sentences were modified to get a more fluid text.

   - We have used both "behavior" and "behaviour" words. In order to have a more consistent article we have replaced "behavior" by "behaviour" in the text. Similarly, "centre" by "center".

   - In addition, the expression "ozone concentration" was replaced by "Total ozone column (TOC)".

   - The word "lines" was replaced by "bands" to indicate the wavelengths where the brewer measured.

**References**

New references have been included in the text where you and the other referees have indicated. However, we think that introducing a reference in the abstract as you suggest in your report is not habitual.

**Abstract**

**P1- L9**. This sentence was rewritten taking into account your suggestions. Also, the standard deviation symbol $\sigma$ has been introduced together with the values reported.

**Introduction**

**P2- L5**. The sentence where we suggest that "... several countries agreed to reduce the agents that produces the decrease of ..." was modified and, the number of countries agreed (197) was included.

**P2- L12**. The expression "Brewer is a spectrophotometer" was replaced by "Brewer ozone spectrophotometer". Also, this paragraph were rewritten.

**P2- L21**. 70 degrees as limit for single Brewer observations seems small, as it corresponds to -values of smaller than 3. We think that is better to say that " The single Brewer presents this problem for large ozone slant column (OSC)".

**P2- L32**. We think that is correct to duplicate the sentence "Since November 2003 and within the World Meteorological Organization (WMO) and the Global Atmosphere Watch ...." because in this paragraph it is used to talk about the history of the RBCC-E and, obviously, you must say: when was created it?, when the brewers were installed ?, which is our travelling Brewer?, etc.

**P3- L11** We say that Izaña has "clean air and clear sky conditions around all the year and offers excellent conditions to perform the Langley-technique". As you indicate we have not taken into account the dust intrusions. Therefore, the sentence was rewritten: "This ensures clean air and clear sky conditions around all the year and offers excellent conditions to perform the Langley technique, except for some days where the Saharan dust intrusions make difficult to measure the direct solar radiation".

**P3- L12**. The sentence "Moreover, comparisons with the World Triad reference are carried out regularly..." was rewritten as " Moreover, the traceability between the RBCC-E Triad and the World Triad Reference is checked during the calibration campaigns through the travelling references #185 and #017."

**P3- L19**. The expression "annually alternating" was replaced by "annually, alternating"

**P3- L23**. The sentence "... on the measurements performed by each Brewer in its local station" was rewritten.

**P4- L4**. The sentence "... the precision between the measurements" was rewritten as "how similar are the measurements made by the Brewers "

**Theoretical Approach**

All referees have indicated that the section 2 is confused. Therefore, it has been rewritten taking into account all the suggestions about the approach to ozone retrieval (DS routine, slits) and Langley-technique. Also, a new figure has been introduced and the equations were modified, adding the symbol $\mu$. In addiction, a clearer explanation about the procedure of the Langley technique was written. In it, we relate the Langley with the inverse of $\mu$, and also explain how it affects the aerosols in this technique.

Now, we hope that the order of the section will be better.

**Ozone and dataset selected**

**P8- L6**. The sentence "Despite this annual behaviour, the ozone is stable during the day, with a low standard deviation for the recorded data." was rewritten as "Despite this annual behaviour, the ozone presents a lower daily variability as indicated by our measurements".

**P8- L17**. The criterion lower than 0.6 under item 4 is not clear: standard deviation of 0.6 DU? Seems to be a very low standard deviation for a day with small ozone variation.

Yes, this condition was introduced in the article by mistake. It was written in a previous version (internal draft) but it was not deleted from the uploaded article.

**Results and discussion**

**P10- L8/9. what does slight mechanical miscalculation mean?**

Sorry, maybe this sentence is not the best translation into English. The text has been modified as follows:

Old version: "... it should be noted that the presence of slight mechanical miscalculations in the instrument...."

New Version: "... it should be noted that the presence of small drifts by its continued operation of the instrument...."

**P12 -L12**. The equation "mean value" was correct.

**P10- L15**. The expression "our experience suggest" was rewritten as "our experience suggests".

**Conclusions**

Some sentences were rewritten. The standard deviation symbol $\sigma$ has been introduced to identify the values reported and a new table where all values are summarised is also included.

[revised manuscript text omitted]

The present work [..[65] ]focused on investigating how similar are the measurements made by the Brewers #157, #183 and #185 [..[66] ]each day and how stable does this behaviour remain over time. This allows us to identify [..[67] ]periods with lower or higher agreement between the Brewers. The RBCC-E measurements are evaluated from the methods described for the World Reference and Arosa Triads to study its stability. With this idea in mind, this work has been structured as follows: an approach to ozone retrieval and Langley method is presented in Section 2. The [..[68] ]ozone values recorded in the period 2005-2016 and datasets used are shown in Section 3. The methods used to calculate the daily ozone value [..[69] ]and the results obtained from these values and its discussion are presented in Section 4. Also, this section incluides results of a study on the behaviour of the RBCC-E Triad as a function of SZA range at which the measurements were performed. Finally, the conclusions are presented in Section 5.
* * *
[61]removed: concentrations

[62]removed: (Redondas et al., 2015; Redondas and Rodriguez-Franco, 2016; De La Casinière et al., 2005)

[63]removed: to calculate the TOC concentration

[64]removed: around

[65]removed: shows the consistence between the measurements performed at IZO

[66]removed: in the period 2005-2016

[67]removed: the years with high or low long-term stability. Also, the accuracy in function of the solar zenith angle (SZA) is studied to evaluate the intraday dependence (short-term stability). This

[68]removed: selection criteria for the datasets evaluated and the procedure

[69]removed: from these are shown in Section 3. The results

[Figure]

**Figure 1.** Ozone and sulfur dioxide absorption cross sections. The solar radiation is measured for the intensity bands ($\lambda_{1-5} = 306.4, 310.1, 313.5, 316.8, 320.0 nm$). In contrast, the wavelength $\lambda_0 = 303 nm$ is used for a check routine.

**2 [..[70] ]**

**2 Theoretical Approach**

**2.1 Ozone retrieval.**

[..[71] ]

5     The standard (so-called DS) routine used to determine the TOC from direct sunlight radiation, measuring the signal intensity in five bands ($\lambda_{1-5} = 306.4, 310.1, 313.5, 316.8, 320.0 nm$) which are associated with maximum and minimum [..[72] ]$O_3$ and $SO_2$ absorption cross sections, see Fig. 1. Despite that $SO_2$ presents a more efficient absorption, its lower presence in the atmosphere (5 D.U.) compared to the ozone (200-500 D.U.) causes that the greater absorption of UV
* * *
[70]removed: Theoretical Approach.

[71]removed: Each Brewer, in its movement following the Sun, measures the direct solar radiationin four spectral lines

[72]removed: ozone absorption bands. The line intensity

radiation is due to the latter (Kerr, 2010). The intensity measured $F$[..[73] ],in raw counts for each wavelength, can be expressed in terms of counts per second, after applying some instrumental corrections [..[74] ](dark counts, dead time [..[75] ]and temperature coefficients) and, also, taking into account the contribution of [..[76] ]Rayleigh scattering. [..[77] ]

[..[78] ]

5    [..[79]]

[..[80] ]Using standard Brewer operational variables, [..[81] ]the TOC can be obtained as follows,

$$O_3 = [..^{82}]\frac{MS9 - ETC}{\alpha \cdot \mu} \tag{1}$$

where [..[83] ]MS9 [..[84] ](so-called double ratio) is calculated as follows, (Brewer, 1973; Kerr et al., 1981, 1985; Kipp & Zonen, 2008).

10    $$MS9 = 10^4 \sum_i w_i \log F_i = 10^4 (\log F_2 - 0.5 \log F_3 - 2.2 \log F_4 + 1.7 \log F_5) \tag{2}$$

The ozone absorption coefficient, $\alpha$, [..[85] ]

[..[86]]

15    [..[87]]

[..[88]]

[..[89] ]is calculated from dispersion test (Redondas et al., 2014a) [..[90] ]

$$\alpha = 10^4 \sum_i w_i \log \alpha_i = 10^4 (\log \alpha_2 - 0.5 \log \alpha_3 - 2.2 \log \alpha_4 + 1.7 \log \alpha_5) \tag{3}$$
* * *
[73]removed: )

[74]removed: on the raw counts (the so called

[75]removed: , and temperature corrections) andalso

[76]removed: the

[revised manuscript text omitted]

[118]removed: , when the ETC values obtained in

[119]removed: its standard deviation is

[120]removed: ETC introduced in the

[121]removed: ETC mean.

[122]removed: ozone concentration, the ETC

[123]removed: So, in a period with a stable behavior, the ETC

[124]removed: increasing

[125]removed: losses its calibration

[126]removed: On

[127]removed: ETC

[128]removed: RBCC-E Triad stability: Ozone and dataset selection criteria.

[129]removed: Representative value of the Total ozone column.

[130]removed: To our knowledge, there are only a few publications where the stability of the World Reference and Arosa Triads is analyzed.In these articles, and due to the large number of ozone measurements performed throughout day, the authors have calculated a representative value of all of them and, from it, the long-term stability of its triad has been analyzed (Fioletov et al., 2005; Stübi et al., 2017; Scarnato et al., 2010).

[131]removed: Fioletov et al. (2005) studied the long-term stability of the World Triad Reference in the period 1985 – 2003. In this work, the authors proposed to fit the measurements performed by each Brewer (

[132]removed: 008, #014 and #015) to a $2^{nd}$ grade polynomial:

[..$^{134}$ ]

[..$^{135}$ ]183 during the year 2011. The vertical lines represent situations which can produce a change in the behaviour of the instrument, while the horizontal line represents the operative $ETC$ used to calculate the TOC (Eq.1). As it can be observed, the ETC changed twice, the first time by maintenance tasks (performed by IOS service in July 2011), and the second time due to changes in the Brewer configuration (to be more precise, changes in the so-called "Cal-Step" in August 2011). On the contrary, during the maintenance tasks (June 2011) or after UV calibration in our facilities (November 2011), the [..$^{136}$ ]

[..$^{137}$]

[..$^{138}$ ]ETC remained constant. Only the Langleys that satisfy the conditions indicated in Sect. 2.2 are used to calculate the weekly mean. Other examples of events that may affect the ETC can be found in the calibration campaign reports (Redondas et al., 2015; Redondas and Rodriguez-Franco, 2016).

[..$^{139}$ ]

[..$^{140}$ ]When a new $ETC$ is given, the [..$^{141}$ ]TOC calculated from it can be compared with the data obtained by other instrument with similar precision. This can be a [..$^{142}$ ]strategy to check if the new ETC is correct. At the RBCC-E Triad, this task is simple because the Brewers are constantly compared to each other, allowing to identify the exact moment when a Brewer needs a new calibration. In addition, the traceability between the RBCC-E and the [..$^{143}$ ]

[..$^{144}$][..$^{145}$][..$^{146}$]

[..$^{147}$ ]
* * *
$^{134}$removed: where $O_3$ are the ozone concentrations measured and $t - t_0$ corresponds to the difference between the time of the measurement and the solar noon. The independent coefficient $A$ obtained through the adjustment is used as a representative ozone value of each instrument. The difference between this coefficient for each Brewer and the Triad mean represents the drifts of each instrument.The stability is studied from daily and monthly mean of these differences.

$^{135}$removed: Stübi et al. (2017) studied the long-term stability of the Arosa Triad in the period 1988 – 2015. In this study,

$^{136}$removed: authors considered that the Triad is the most appropriate reference for each day. Therefore, the measurements of the three Brewers are modeled as a 3$^{rd}$ grade polynomial dependent on time :

$^{138}$removed: where $t_0$ corresponds to the 12 UTC time. In this case, each Brewer is characterized by a shift $\Delta$, which is the mean of the difference between the values measured and obtained from the fit, and a standard deviation $\sigma$. The standard deviation $\sigma$ evaluates the dispersion of these differences. Both parameters are used to analyzed the long-term stability of the Arosa Triad.

$^{139}$removed: In order to compare the long-term stability of the RBCC-E Triad with respect to the World Reference and Arosa Triads, both expressions are used to fit our measurements in this work.In this work, the time reference $t_0$ is the solar noon.

[revised manuscript text omitted]

[192]removed: The consistence of the measurements carried out by the RBCC-E Triadwas evaluated from the methods and datasets described in Sect. 3. In the case of the long-term stability, it was studied using both datasets, while the short-term stability was analyzed using only Dataset 1. The results obtained are shown in statistical terms.

[193]removed: .

[199]removed: triad

[200]removed: grade polynomial. In addition, in this work the results using a

[201]removed: have also been investigated

[202]removed: A value

[203]removed: daily difference calculate from daily mean, $A_M$, of all measurements is included.

[204]removed: in contrast to the mean

[205]removed: at tropical latitudes. Therefore,

[206]removed: A coefficients. Therefore, the

[207]removed: of the measurements made by the Brewers

[208]removed: the selected method

[209]removed: . This result suggests that the Brewers of the RBCC-E Triad are in

[210]removed: as daily data that the standard deviation mean of the Canadian Triad

[211]removed: is the average

[212]removed: deviations of each Brewer

[Figure]

**Figure 5.** Daily difference of the ozone reference value [..[194] ]$A$ of each [..[195] ]Brewer with respect to Triad[..[196] ]. The [..[197] ]values were obtained from the procedure proposed by Fioletov (World Triad Reference) and by daily [..[198] ]mean (RBCC-E).

Table 3 contains the difference mean, calculated from the mean Brewer-Triad difference plotted in Fig. 5, and its standard deviation. The RBCC-E Triad presents a relative standard deviation mean equal to 0.41% ($\sigma_{157} = 0.362\%$, $\sigma_{183} = 0.453\%$ and $\sigma_{185} = 0.428\%$; see Table 3, Dataset 1, [..[213] ]$4^{th}$ column). This result indicates that the dispersion of the measurements of the
* * *
[213]removed: column $A_2(\%)$

**Table 3.** Absolute and relative values of the mean shift and the standard deviation.

|  | Dataset 1 | | | | | |
|---|---|---|---|---|---|---|
|  | [..216 ]2nd grade polynomial (DU) | 3rd grade [..217 ]polynomial (DU) | Brewer [..218 ]Mean (DU) | 2nd grade [..219 ]polynomial (%) | 3rd grade [..220 ]polynomial (%) | Brewer [..221 ]Mean ( |
| Brewer #157 | $0.787 \pm 1.04$ | $0.796 \pm 1.07$ | $0.569 \pm 0.744$ | $0.276 \pm 0.362$ | $0.279 \pm 0.372$ | $0.1994 \pm 0.$ |
| Brewer #183 | $0.989 \pm 1.29$ | $1.01 \pm 1.31$ | $0.741 \pm 0.956$ | [..222 ]$0.349 \pm 0.453$ | [..223 ]$0.356 \pm 0.463$ | [..224 ]$0.26 \pm 0$ |
| Brewer #185 | $0.89 \pm 1.21$ | $0.90 \pm 1.23$ | $0.56 \pm 0.78$ | [..225 ]$0.315 \pm 0.428$ | [..226 ]$0.32 \pm 0.438$ | [..227 ]$0.20 \pm 0$ |

|  | Dataset 2 | | | | | |
|---|---|---|---|---|---|---|
|  | [..228 ]2nd grade polynomial (DU) | 3rd grade [..229 ]polynomial (DU) | Brewer [..230 ]Mean (DU) | 2nd grade [..231 ]polynomial (%) | 3rd grade [..232 ]polynomial (%) | Brewer [..233 ]Mean ( |
| Brewer #157 | $0.795 \pm 1.00$ | [..234 ]$0.82 \pm 1.05$ | $0.534 \pm 0.661$ | $0.278 \pm 0.349$ | $0.286 \pm 0.368$ | $0.186 \pm 0.$ |
| Brewer #183 | $0.747 \pm 0.942$ | $0.784 \pm 1.02$ | $0.547 \pm 0.719$ | [..235 ]$0.262 \pm 0.331$ | [..236 ]$0.275 \pm 0.36$ | [..237 ]$0.192 \pm 0$ |
| Brewer #185 | $0.64 \pm 0.87$ | $0.67 \pm 0.99$ | $0.376 \pm 0.526$ | [..238 ]$0.227 \pm 0.311$ | [..239 ]$0.238 \pm 0.333$ | [..240 ]$0.133 \pm 0$ |

RBCC-E Brewers presents [..214 ]a similar behaviour to those of the [..215 ]World Triad Reference. Furthermore, the standard deviation values obtained confirm that the daily mean is the best method to evaluate the RBCC-E Triad.

In order to compare the daily behaviour of the Arosa and RBCC-E Triads, a 3rd grade polynomial was fitted to all the daily measurements made by [..241 ]RBCC-E Brewers for Datasets 1 and 2. Then, for each Brewer its mean shift, $\Delta$, and standard

5  deviation, $\sigma$, were calculated [..242 ](see Sect. 3). The values obtained for the Dataset 1 are shown in Fig. 6. Because Brewer #183 was damaged by a storm and was inoperative between December 2005 and September 2006, the data plotted in that period were calculated from measurements of Brewers #157 and #185 only. Similarly, when Brewer #185 is away from IZO in calibration campaigns, the values plotted correspond to Brewers #157 and #183. Note that although these data were introduced in Fig. 6 to avoid gaps in the plot, they are not considered in the statistical study. Therefore, the dates evaluated correspond

10  with the days when the full RBCC-E Triad is operative, and the criteria established in Sect. 3 are still used.

As [..243 ]can be observed in Fig. 6, the results obtained for all instruments in Dataset 1 show a ($\pm 0.5$) value for the mean shift. A similar result was obtained for Dataset 2[..244 ], figure not shown[..245 ]. Contrary to the report in Stübi et al. (2017), in the present case the standard deviation does not show any seasonal component. Again, this result is explained by the [..246 ]low
* * *
[214]removed: the same behaviour as
[215]removed: Canadian Triad
[241]removed: the
[242]removed: (Stübi et al., 2017).
[243]removed: it
[244]removed: (
[245]removed: )
[246]removed: almost constant value

[Figure]

**Figure 6.** Time series of the mean shift $\Delta$ and the standard deviation $\sigma$, in terms relative to the TOC calculated from measurements performed by the three Brewers of the RBCC-E Triad (#157, #183 and #185) fitted with a $3^{rd}$ grade polynomial

daily variability of the ozone at [..247 ]sub-tropical latitudes. For the Brewers of the RBCC-E Triad, the standard deviation is more influenced by any anomalous internal [..248 ]behaviour of the instruments. For middle latitudes, e.g. in Arosa, there is a larger daily variation in ozone and the standard deviation shows it.

Following Stübi et al. (2017), Table 4 shows the distribution of percentiles of the mean shift and the standard deviation
5   values plotted in Fig. 6. [..249 ]The Brewers present a similar interpercentile range $P_{2.5} - P_{97.5}$, with a mean value close to 1.1%. This result is consistent with the standard deviation shown in Table 3 for the polynomial fits[..250 ]. In comparison with the Arosa Triad, only Brewer #040 shows a better [..251 ]behaviour than the RBCC-E Brewers [..252 ]while their other Brewers (B#072, B#156) show similar values to [..253 ]ours.

**4.1.2   Long-term stability: monthly averages**

10  Although the histogram and the statistical parameters already presented suggest that the long-term stability of the RBCC-E[..334 ], Arosa and World Reference Triads are similar. It can be more interesting to [..335 ]present this study from the
* * *
247 removed: tropical

248 removed: behavior

249 removed: All

250 removed: , where the long-term stability of the Brewer-Triad mean was studied by the procedure proposed by Fioletov et al. (2005)

251 removed: behavior

252 removed: . The other two Brewers of the Arosa Triad

253 removed: those of the RBCC-E

334 removed: Triad is similar to that of the World Reference and Arosa Triads , it

335 removed: study the stability using

**Table 4.** Percentiles of the difference distribution (%) for the RBCC-E Triad.

[revised manuscript text omitted]

---

## Referee Report (RR1)

Review of Stability of the Regional Brewer Calibration Center for Europe
Triad during the period 2005 – 2016
By Sergio Fabián León-Luis et al

V.Savastiouk.

General comments:
   The revised text shows significant improvements compared to the originally submitted. The revised manuscript still reads a little too much as popular science with unconventional terminology and loosely used words ("measuring the signal intensity" or "The grating system separates the solar radiation").  If the reader overlooks this then it is possible to understand the significance of the good long-term agreement of the RBCC-E triad of Brewers.

Title:   The word stability in the title implies that the Brewer triad at RBCC-E was stable, but this is not what the content of the paper shows. It shows good agreement between the three Brewers that were regularly absolutely and independently calibrated, or re-calibrated.  The paper doesn't claim or show that the triad was not changing, however it does show that the changes did not affect the agreement between the Brewers once they have been properly calibrated.  I suggest using something like "consistency" instead of "stability".

P1L4   "its own calibration" may be interpreted as "different" from others. I suggest saying that RBCC-E is using traveling standard(s) that are absolutely and independently calibrated at Izana.

P1L19 replace "measurement" with "calculation"

P1L20 "accuracy" implies that you are comparing to the true value of some sort. What you are investigating is sensitivity or precision (or uncertainty)

P2L1-3  these sentences are not connected to each other. It looks like you've removed some text and didn't rephrase what's left.  E.g. what "decrease"?

P2L10 "This instrument is mounted on an azimuth tracker that determines the TOC..."  the tracker definitely does not determine the TOC!

P2L11  "The grating system separates the solar radiation"  separates into/from what? Please use the established scientific terminology. "The grating system" as you called it is in fact a diffraction grating that can be turned with a stepper motor that together with a slit mask allows the selection of solar radiation bands at the exit slit(s) of the monochromator.

P2L17  Stray light certainly doesn't cause any decrease in TOC. It causes underestimation of TOC in the calculations when no stray light correction is applied.
P2L32 "its own" see comment above
P3L16 "conducted"

P5EQ1 what are the units of O3 in this equation?  Be careful, this is a trick question (but important!).

---

## Author Response (AR2)

**Internal consistency of the Regional Brewer Calibration Center for Europe Triad during the period 2005 – 2016**

Sergio Fabián León-Luis1,2, Alberto Redondas1,2, Virgilio Carreño1,2, Javier López-Solano1,2,3, Alberto Berjón2,3, Bentorey Hernández-Cruz1,2,3, and Daniel Santana-Díaz2,3

1Izaña Atmospheric Research Center, Agencia Estatal de Meteorología, Tenerife, Spain

2Regional Brewer Calibration Center for Europe, Izaña Atmospheric Research Center, Tenerife, Spain

3Departamento de Ingeniería Industrial, Universidad de La Laguna, Tenerife, Spain

Correspondence to: Alberto Redondas (aredondasm@aemet.es)

**General Comments:**

**Comment**: My first impression reading the first pages of the manuscript was, that its rewriting significantly improves not only the English but also the content and structure. All addressed issues in the first review(s) have been incorporated. Abstract and introduction are now in an acceptable form. However, coming to chapter 2, 3 and 4 more and more still unsatisfying issues showed up. Thus I am not able to recommend now its publication without further major revision.

Description of the data retrieval and Langley calibration method (chapter 2) and of the selected data sets (chapter 3) is still not perfect, but better than in the first version and more or less acceptable with some minor corrections. Chapter 4 is still difficult to read and it is hard to understand the presented findings and conclusions. Some sentences are confusing, some statements do not reflect the results (e.g. p13, 15 and figure: no seasonal component in the relative standard deviation is in my opinion a very optimistic statement).

5

Answer: The authors are grateful for the comments of the referee and his description about the problems that he has found in each chapter. To solve them, we have improved the wording of some paragraphs in the text.

**Comment:** In addition results of the regular standard lamp tests of the Brewers are missing, which are normally a good indicator of the consistency of the instrumental calibration (sorry that I did not mention this issue in my first review). A

15

comparison of these measured SL-test records with the presented statistical parameters should be included and hopefully show the same good consistency.

Answer: The results of the SL-test will not be included in the article but as supplementary information. We think (as the editor) that including them in the text can lead to a more complex article where the reader can get lost more easily. Moreover, at the RBCC-E, the SL-test is not the best indicator of the status of a Brewer (they are a secondary indicator for us). This is due to there are changes on the characteristics of the instrument that can be not detected with a SL-test such as variations on the attenuation filters, iris and pointing. The importance of having established the Regional

20

Brewer Calibration Center in the Izaña Atmospheric Observatory is because this place presents the ideal conditions for the Langley technique which allows us an absolute and independent derive the Extraterrestrial constant (ETC) of each brewer. These ETC values calculated from Langley are used to identify the possible changes on the spectral sensitivity of our instruments.

- **Comment:** Chapter 5 Conclusion is a little on the short side and especially the statement in the last sentence is based mainly 5 on one mention of the 0.5%-agreement of BR17 and BR185 (P3, 112) during some campaigns. The results in the tables (especially table 6) only show the internal consistency within the Brewers of the various triads and that their long term consistency is good. But it does not say anything about their accuracy and the comparability of the triads' calibration levels, which is a condition for the traceability of the ozone measurements all around the world, when different triads are used for calibration of the field instruments in the global network.
- 10

**Answer:** The compatibility of the Triads is a pending topic due the lack of intercomparisons between triads, we can only address by indirect intercomparisons with BR17, the conclusion has been rewritten accordingly.

**Specific Comments (recommended text in "text")**

- P2, 110: This instrument is mounted on an azimuth tracker "and instead that" determines the TOC...... The wording of this

**15 sentence has been changed**

- P2, 113 - 14: better "1985), it has been subjected on-going technical improvements to enhance its accuracy". The wording

**of this sentence has been changed**

- P2, 114 – 15: no complete sentence. The wording of this sentence has been changed

- P2, 128: replace manufacturer by with "by manufacturer". done

- P3, 13: "is used for developing and tests". done 20
  - P3, 112: "in these comparisons". done

- P3, 113: what kind of range? SZA!Not, in this point we reference to the ozone slant column (OSC). The wording of this sentence has been changed

- P3, 117/18: "Moreover, in order to obtain ozone values with higher accuracy, the RBCC-E advises on the need to reprocess

the preceding observation records of each Brewer at its local station". done 25

- P3, 120/21: no complete sentence. Perhaps it would be nice to mention ATMOZ already here in the context of Absolute Calibration Campaigns of the Brewer and Dobson reference instruments. Now, in the introduction, we reference to this

**project and about the Absolute Calibration Campaigns of the Brewer and Dobson reference instruments**

- P3, 133 34: not a good English sentence. The wording of this sentence has been changed
- 30 - P4, 14: "includes" done
  - P4, 111: "< 5 D.U.". done
  - P5, 11: "the measured intensities Fi in raw counts for each wavelength i". done

- P5, 124: The Langley technique is most popular only for reference instruments' calibration, but not for field instruments.

**The wording was changed to indicate that only the reference instrument used this calibration technique**

- P5 – P6 general statement: description is still not perfect, but better than the first version, therefore generally acceptable. We have written both pages, a now a better description is given.

5 - P6, 114: (2 Langley "plots" per day). **done**

- P6, 115 ff: Meaning of bullet point 3 is not clear. Normally a regular Brewer TOC observations is calculated as average out of 5 single measurements and can be used if the standard deviation is less than 2.5 D.U. What is meant with individual measurements? These do not have a standard deviation. P6, 117 ff: Bullet point 4: Misleading formulation, as this daily standard deviation limit of 2.5 D.U. does not come from the description under bullet point 3, which refers to individual observations.

**10 We have rewritten the point 3 where the cloud screening filter and the single measurements are introduced. The point 4 has been deleted**

- P6, 124/25: Wording Only when the instruments recalibrates is uncommon. I would say "changing only when the calibration of the instrument drifts". **done**

- General comment on the above discussed section: Here it would be proper to mention the SL-test as a simple alternative to

15 monitor the instrument's calibration level. We have prepared an additional document with the SL and Langley values of each instruments during the period 2005-2016.

- P7, Figure 2: Error bars hardly discernible, quality of the graph is not optimal. A new Figure, with better resolution, has been added.

- P8, 14: add "Saharan" dust done

20

- P8, 115 ff: Construction of this sentence is not good. The wording of this sentence has been changed

- P11, 15: "daily" mean? done

- P11, 115 - 23: Sorry but I have problems to understand the conclusion. Somewhat confusing! We have rewritten the conclusion of this paragraph

P12, Figure 5: more detailed caption needed. Perhaps it helps to assign the different procedures to the shown panels. For a
better comparability it will be helpful to use the same y-axis maximum in all sub-graphs. A new Figure, with the same Y-axis, has been uploaded

- P13, 15/figure 6on P14: It is stated there is no seasonal component in the standard deviation. This IMHO not the reality, as I can see a seasonal variation. Yes a small seasonal component in the Std can be identified. The sentence has been rewritten as "Similarly to report by Stübi et al. (2017) the standard deviation shows an small seasonal component."

30

- P13-14, 110, table 4: The mentioned 1.1% in table 4 is not clear. How is it determined? I see only numbers less than +-1%. It is not possible for me follow the conclusion of consistency **This number corresponds to the sum of the percentile values**,  $P_{2.5} + P_{97.5}$ , in absolute value (see Table 4). this factor was used by Stübi et al. (2017) to describe the behaviour of the Arosa triad.

- P13 & 14 - 15: An explanation for the higher Arosa standard deviation is given, but isn't it the same reason for the Toronto 35 standard deviation? Honestly, this explanation we believe that only valid to explain why the RBCC-E presents a smaller deviation than the Arosa Triad, using its procedure to study the consistency of its measurements. The Toronto Triad used another procedure to study the consistency of their measurements and, hence, the standard deviation is calculated differently. Moreover, this triad is formed by single Brewers and each instrument understimates the TOC for large SZA differently. In this case, we don't know how affect this on the standard deviation, Fioletov et al. (2005) only indicated the daily value of this

5 parameter but we can not find more informationfigures, tables, etc. In constant, the the RBCC-E uses double brewer which do not need stray light correction and, hence, the standard deviation only depends on the daily variability of the ozone at sub-tropical latitudes.

- P15, Figure 7: in figure caption please use the plural "methods" and "its micrometers" instead of their micrometers. done

- P15, 13: Be more precise, as it is 36% lower than 40%. The sentence was rewritten as "The ratio between 3-monthly

**10 standard deviation is 36% lower for the RBCC-E"**

- P16, l4: "allows to" done

- P16 – 17: What about the afternoon drop in figure 8, left panel? It is not explained. In this section "Short-term consistency", we study the consistency of our data as a function of the SZA used the Dataset 2 (only simultaneous measurements between 2010-2016). The drop observed in the Figure 8 is due to that one instrument was not operative during

15 few minutes and, hence, we do not have simultaneous measurements between the Brewers. This information have been written in the Figure Caption.

- P17: As already mentioned under general comments the conclusion is a little bit short. The conclusion has been rewritten

- P21, references: please write "Köhler" and "Kyrö". done

Sergio Fabián León-Luis1,2, Alberto Redondas1,2, Virgilio Carreño1,2, Javier López-Solano1,2,3, Alberto Berjón2,3, Bentorey Hernández-Cruz1,2,3, and Daniel Santana-Díaz2,3

1Izaña Atmospheric Research Center, Agencia Estatal de Meteorología, Tenerife, Spain

2Regional Brewer Calibration Center for Europe, Izaña Atmospheric Research Center, Tenerife, Spain

3Departamento de Ingeniería Industrial, Universidad de La Laguna, Tenerife, Spain

Correspondence to: Alberto Redondas (aredondasm@aemet.es)

**General comments**

5

10

- Comment: The revised text shows significant improvements compared to the originally submitted. The revised manuscript still reads a little too much as popular science with unconventional terminology and loosely used words ("measuring the signal intensity" or "The grating system separates the solar radiation"). If the reader overlooks this then it is possible to
- understand the significance of the good long-term agreement of the RBCC-E triad of Brewers. Answer: We have rewritten these phrases to get a text with a better scientific language.
- **Comment:** Title: The word stability in the title implies that the Brewer triad at RBCC-E was stable, but this is not what the content of the paper shows. It shows good agreement between the three Brewers that were regularly absolutely and independently calibrated, or re-calibrated. The paper doesn't claim or show that the triad was not changing, however it does show that the changes did not affect the agreement between the Brewers once they have been properly calibrated. I suggest using something like "consistency" instead of "stability".

Answer: The editor and you agree that the word "stability " is not the best to describe the objective of this article. Therefore, we have changed the title of "Stability of the Regional..." by "Internal consistency of the regional...". In text, we think that is better used only "consistency" to speak about the agreement of our measurements..

15 Comment: P1L4 "its own calibration" may be interpreted as "different" from others. I suggest saying that RBCC-E is using traveling standard(s) that are absolutely and independently calibrated at Izaña.

Answer: We rewritten this sentence as "the RBCC-E transfers its calibration based on Langley using travelling standard(s) that are absolutely and independently calibrated at Izaña".

1

Comment: P1L19 replace "measurement" with "calculation"

20 Answer: The text has been modified following your suggestion

Comment: P1L20 "accuracy" implies that you are comparing to the true value of some sort. What you are investigating is sensitivity or precision (or uncertainty)

Answer: The text has been modified following your suggestion

Comment: P2L1-3 these sentences are not connected to each other. It looks like you've removed some text and didn't rephrase

5

what's left. E.g. what "decrease"?

Answer: We have modified the paragraph and, now, the wording is better.

Comment: P2L10 "This instrument is mounted on an azimuth tracker that determines the TOC..." the tracker definitely does not determine the TOC!
Answer: We have modified this paragraph and, now, the wording is better.

10 Comment: P2L11 "The grating system separates the solar radiation" separates into/from what? Please use the established scientific terminology. "The grating system" as you called it is in fact a diffraction grating that can be turned with a stepper motor that together with a slit mask allows the selection of solar radiation bands at the exit slit(s) of the monochromator. Answer: The wording of this paragraph was changed Comment: P2L17 Stray light certainly doesn't cause any decrease

in TOC. It causes underestimation of TOC in the calculations when no stray light correction is applied.

**15 **Answer:** The text has been modified following your suggestion**

Comment: P2L32 "its own" see comment above

Answer: The text has been modified following your suggestion

Comment: P3L16 "conducted"

Answer: The text has been modified following your suggestion

20 Comment: P5EQ1 what are the units of O3 in this equation? Be careful, this is a trick question (but important!). Answer: We have rewritten the equations. Now, the TOC is given in Dobson units.

**[..\* ]Internal consistency of the Regional Brewer Calibration Center for Europe Triad during the period 2005 – 2016**

Sergio Fabián León-Luis1,2, Alberto Redondas1,2, Virgilio Carreño1,2, Javier López-Solano1,2,3, Alberto Berjón2,3, Bentorey Hernández-Cruz1,2,3, and Daniel Santana-Díaz2,3

[revised manuscript text omitted]

<sup>6removed: stability

<sup>7removed: accuracy

<sup>8removed: Concerns related to the

<sup>9removed: (Basher, 1985; Varotsos and Cracknell, 1994; Fioletov et al., 2005; Scarnato et al., 2009)

<sup>10removed: is mounted on an azimuth tracker that

<sup>11removed: The grating system separates the solar radiation and a slit mask mechanism is used to select the different UV

<sup>12removed: which are associated with maximum and minimum ozone absorption cross-sections

<sup>13removed: had

<sup>14removed: improve

<sup>15removed:,

<sup>16removed: (Fioletov et al., 2005, 2011; Karppinen et al., 2015; Fountoulakis et al., 2017)

<sup>17removed: a decrease

[revised manuscript text omitted]

The present work focused on investigating how similar are the measurements made by the Brewers #157, #183 and #185 each day and how stable does this behaviour remain over time. This allows us to identify periods with lower or higher agreement between the Brewers. The RBCC-E measurements are evaluated from the methods described for the World Reference and Arosa Triads to study its  $[...^{28}]$  consistency. With this idea in mind, this work has been structured as follows: an approach to

25

ozone retrieval and Langley method is presented in Section 2. The ozone values recorded in the period 2005-2016 and datasets used are shown in Section 3. The methods used to calculate the daily ozone value and the results obtained from these values and its discussion are presented in Section 4. Also, this section  $\left[ ...^{29} \right]$  includes results of a study on the behaviour of the RBCC-E Triad as a function of SZA range at which the measurements were performed. Finally, the conclusions are presented in Section

5.

<sup>24removed: conduted

<sup>25removed: an ozone value with better

<sup>26removed: observations performed by

[revised manuscript text omitted]

<sup>47removed: needs

<sup>48removed: the